# Complex pattern of facial remapping in somatosensory cortex following congenital but not acquired hand loss

Victoria Root[1,2,3†], Dollyane Muret[2*†], Maite Arribas[2,4], Elena Amoruso[2,3], John Thornton[5], Aurelie Tarall-Jozwiak[6], Irene Tracey[1], Tamar R Makin[2,3,5]

[1]WIN Centre, University of Oxford, Oxford, United Kingdom; [2]Institute of Cognitive Neuroscience, University College London, London, United Kingdom; [3]Medical Research Council Cognition and Brain Sciences Unit (CBU), University of Cambridge, Cambridge, United Kingdom; [4]Department of Psychosis Studies, Institute of Psychiatry, Psychology & Neuroscience, King's College London, London, United Kingdom; [5]Wellcome Trust Centre for Neuroimaging, University College London, London, United Kingdom; [6]Queen Mary's Hospital, London, United Kingdom

**Abstract** Cortical remapping after hand loss in the primary somatosensory cortex (S1) is thought to be predominantly dictated by cortical proximity, with adjacent body parts remapping into the deprived area. Traditionally, this remapping has been characterised by changes in the lip representation, which is assumed to be the immediate neighbour of the hand based on electrophysiological research in non-human primates. However, the orientation of facial somatotopy in humans is debated, with contrasting work reporting both an inverted and upright topography. We aimed to fill this gap in the S1 homunculus by investigating the topographic organisation of the face. Using both univariate and multivariate approaches we examined the extent of face-to-hand remapping in individuals with a congenital and acquired missing hand (hereafter one-handers and amputees, respectively), relative to two-handed controls. Participants were asked to move different facial parts (forehead, nose, lips, tongue) during functional MRI (fMRI) scanning. We first confirmed an upright face organisation in all three groups, with the upper-face and not the lips bordering the hand area. We further found little evidence for remapping of both forehead and lips in amputees, with no significant relationship to the chronicity of their phantom limb pain (PLP). In contrast, we found converging evidence for a complex pattern of face remapping in congenital one-handers across multiple facial parts, where relative to controls, the location of the cortical neighbour – the forehead – is shown to shift away from the deprived hand area, which is subsequently more activated by the lips and the tongue. Together, our findings demonstrate that the face representation in humans is highly plastic, but that this plasticity is restricted by the developmental stage of input deprivation, rather than cortical proximity.

**\*For correspondence:**
dollyane.muret@inserm.fr

†These authors contributed equally to this work

## Editor's evaluation

This fundamental work substantially advances our understanding of cortical remapping in people with congenital or acquired missing hands. The evidence supporting the idea that remapping may not follow cortical proximity but instead functional rules as to how the effector is used are compelling, with rigorous univariate and multivariate analyses applied to functional Magnetic Resonance Imaging data. Importantly, the authors suggest this is mostly the case for one-handers but not for amputees for who the reorganization seems more limited in general.

## Introduction

Our brains capacity to adapt, known as cortical plasticity, is integral to our successful functioning in daily life, as well as rehabilitation from injury. A key model for exploring the extent, and consequences of, cortical plasticity is upper-limb loss (via amputation or congenital absence). Here, the cortical hand territory in the primary somatosensory cortex (hereafter S1), suffers an extreme loss of sensory input in tandem with dramatic alterations of motor behaviour (*Makin et al., 2013a*; *Muret and Makin, 2021*). The functional and perceptual correlates of amputation-related plasticity are currently debated (*Makin and Bensmaia, 2017*; *Ortiz-Catalan, 2018*). In particular, it is not clear whether functional cortical reorganisation is restricted to early life development or can also occur in adults.

Traditionally, research assessing cortical plasticity after upper-limb loss has followed the tenet that neighbouring body parts of the missing hand, and the lower face in particular, shift and encroach into the deprived hand area. This emphasis on the lip representation stems from early electrophysiological work in non-human primates, where numerous studies demonstrated an 'upside-down' facial somatotopy, with the lower face immediately neighbouring the hand (*Dreyer et al., 1975*; *Merzenich et al., 1978*; *Sur et al., 1982*; *Cusick et al., 1986*; *Lin and Sessle, 1994*; *Manger et al., 1995*; *Manger et al., 1996*; *Jain et al., 2001*; *Cerkevich et al., 2014*). Here, the lips and/or chin inputs have been shown to remap into the deprived hand area after sensory loss (*Pons et al., 1991*; *Jain et al., 1997*), leading to the well-accepted assumption that remapping is determined by cortical proximity (*Buonomano and Merzenich, 1998*; *Nardone et al., 2013*). Thereafter, human measurement of topographic shifts has tended to focus on that of the lips, where researchers have reported that shifted lip representation towards and into the deprived hand area is significantly associated with phantom limb pain (PLP) intensity (*Flor et al., 1995*; *Birbaumer et al., 1997*; *Lotze et al., 2001*; *Grüsser et al., 2001*; *Foell et al., 2014*). PLP is a neuropathic pain syndrome experienced in the missing, amputated limb by the majority of amputees (*Limakatso et al., 2019*). This condition is commonly thought to arise from maladaptive cortical plasticity in S1 (although see *Makin, 2021*), specifically from a signal mismatch between the missing hand representation and the remapped inputs of the lips in the deprived hand area (*Ramachandran and Hirstein, 1998*).

The research focus on lip cortical remapping in amputees is based on the assumption that the lips neighbour the hand representation. However, this assumption goes against the classical upright orientation of the face in S1 (*Penfield, 1950*; *Schwartz et al., 2004*; *Roux et al., 2018*; *Sato et al., 2005*; *Willoughby et al., 2020*), as first depicted in Penfield's Homunculus and in later intracortical recordings and stimulation studies (*Penfield, 1950*; *Schwartz et al., 2004*; *Roux et al., 2018*; *Sato et al., 2005*), with the upper-face (i.e. forehead) bordering the hand area. Furthermore, neuroimaging studies in humans studying face topography provided contradictory evidence for the past 30 years. While a few neuroimaging studies provided partial evidence in support of the traditional upright face organisation (*Willoughby et al., 2020*), other studies supported the inverted (or 'upside-down') somatotopic organisation of the face, similar to that of non-human primates (*Yang et al., 1993*; *Servos et al., 1999*). Other studies suggested a segmental organisation (*Moulton et al., 2009*), or even a lack of somatotopic organisation (*Iannetti et al., 2003*; *Nguyen et al., 2004*; *Kopietz et al., 2009*), whereas many studies provided inconclusive or incomplete results (*Mogilner et al., 1994*; *Hoshiyama et al., 1996*; *Disbrow et al., 2003*; *Nevalainen et al., 2006*). Together, the available evidence does not successfully converge on face topography in humans. In line with the upright organisation originally suggested by Penfield, recent work reported that the shift in the lip representation towards the missing hand in amputees was minimal (*Makin et al., 2015*; *Raffin et al., 2016*), and likely to reside within the face area itself. Surprisingly, there is currently no research that considers the representation of other facial parts, in particular the upper-face (e.g. the forehead), in relation to plasticity or PLP. Detailed mapping of the upper and lower face is therefore needed to assess typical topography of facial sensorimotor organisation, as well as remapping after limb loss.

More recent electrophysiological studies in monkeys demonstrated that much of the face remapping observed in the primary sensorimotor cortex following upper-limb deafferentation does not result from cortico-cortical plasticity, but instead arises from plasticity in the brainstem (*Florence and Kaas, 1995*; *Kambi et al., 2014*). This important finding also highlights the limited explanatory power of local activity increase that has been historically used to infer changes to the representational features of a given brain. Does it reflect a gain modulation of input coming from the brainstem, or increase of local processing of the face input in the hand area? While it remains challenging to dissociate

these two contributions, alternative analysis tools are becoming increasingly popular for mining richer information of the processing underlying activity in a given cortical region. Multivariate analyses are sensitive to more subtle changes in representational content, that are not accessible with the traditional univariate approach. In the context of facial remapping, if the deprived hand area updates its local processing to include facial information (and does not just display more facial activity), we would expect it to show greater information about representational features relevant to different facial parts.

Remapping after upper-limb loss has also been documented in individuals born without a hand (hereafter one-handers), who do not experience PLP (*Makin et al., 2013b*). Here, it has been shown that the representation of multiple body parts, including the residual arm, legs and lips, remapped into the missing hand territory (*Hahamy et al., 2017*; *Hahamy and Makin, 2019*). Importantly, cortical remapping in this group does not depend on cortical proximity. With regard to the lips, a recent transcranial magnetic stimulation (TMS) study has reported functionally relevant lip activity in the deprived hand area of one-handers (*Amoruso et al., 2021*), showcasing that reported remapping may also be functional. It was proposed that the observed remapping of various body parts could have been shaped by compensatory behaviour (*Hahamy et al., 2017*), as these body parts are all used by one-handers to compensate for their missing hand function, but this hypothesis awaits validation.

Here, we conducted a mapping of face cortical organisation to determine facial orientation (upright versus inverted) in the primary sensorimotor cortex in 22 two-handed controls, 17 amputees, and 21 one-handers using an active functional MRI paradigm, where participants were visually instructed to move their forehead, nose, lips or tongue. This paradigm was chosen because it enabled simultaneous bilateral activation of S1 within individual participants, providing a within-participants control design. We also explored the representation of multiple facial movements, which have not been previously studied in the context of deprivation-triggered brain plasticity. To measure the extent of cortical remapping of the upper (forehead) and lower (lips) face in relation to the deprived (or non-dominant) hand area across all groups, we used surface-based topographic comparisons. For this purpose, we

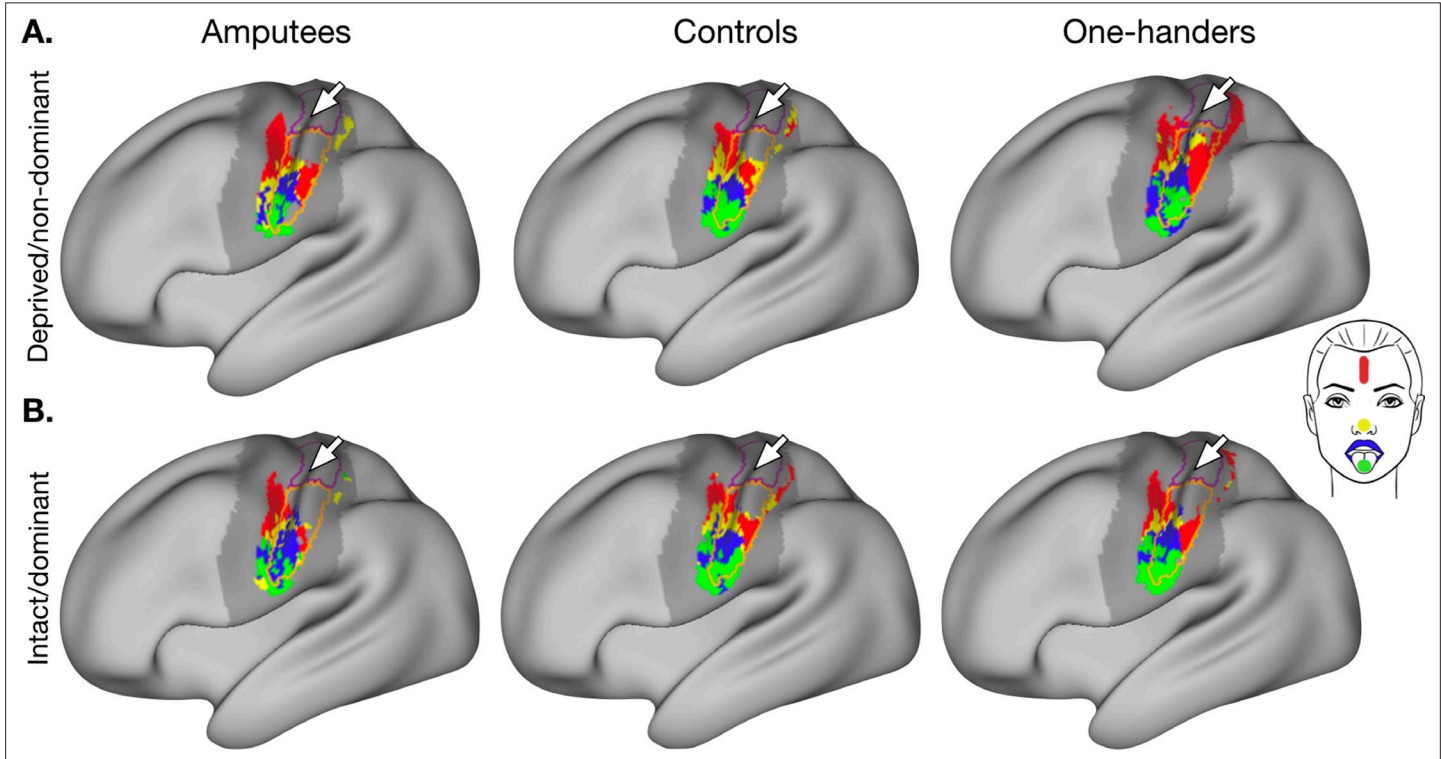

**Figure 1.** Group-level activity maps for each facial movement versus rest. Group average activity for the forehead (red), nose (yellow), lips (blue) and tongue (green) movements, contrasted to rest, in the (**A**) deprived/non-dominant and (**B**) intact/dominant hemisphere of controls (n=22), amputees (n=17), and one-handers (n=21). All clusters were created using a threshold-free cluster enhancement procedure with a sensorimotor pre-threshold mask (defined using the Harvard Cortical Atlas; outlined in darker grey), and thresholded at p<0.01. The hand and face regions of interest (ROIs) are outlined in purple and orange respectively, and the central sulcus is denoted with a white arrow.

employed up-to-date methodology to harvest traditional measures, that is winner-takes-all assessment of surface coverage in the hand and face areas, followed by cortical (geodesic) distances, and similarity analysis of selectivity maps. Furthermore, to go beyond the gross topographic properties of face representation, we used multivariate representational similarity analysis (RSA). This approach allows us to characterise more subtle alterations in the relationship between facial activity patterns (forehead, nose, lips, tongue) in the deprived hand and the face areas.

We found that, in line with Penfield's original description, facial topography was arranged in an upright manner, with the forehead (i.e. upper-face) bordering the hand area across all groups. Contrary to traditional theories (*Flor et al., 2006*), we did not find evidence for facial remapping of either the lips or the (cortically neighbouring) forehead, into the deprived hand area of amputees. We did, however, observe significant remapping of multiple face parts (upper and lower face) in the one-handers group, validating our methodology as suitable for identifying remapping effects. Interestingly, remapping of the cortical neighbour (upper-face) within the one-handers group was away from the missing hand area, while the lips and tongue representations shifted towards the deprived cortex, hinting that the underlying mechanism of remapping is more complex than simple cortical proximity.

## Results

### The cortical neighbour of the hand representation is the forehead

We first visualised the average group activity resulting from active movements with each of the facial parts (versus rest), within a broad sensorimotor mask. When looking at gross facial organisation at the group-level, we found qualitatively similar activity maps across groups (see *Figure 1*), highlighting a robust somatotopy of the face with preserved symmetry across the two hemispheres. These facial maps also indicate an upright orientation of the face in S1, with the forehead located closest to the hand area, followed by the nose, lips, and the tongue located laterally, across all groups. The facial somatotopy presented here therefore suggests that the hand's cortical neighbour is the forehead (or upper-face), highlighting the need to reassess the often-cited, traditional lip-to-hand marker of cortical remapping in amputees and one-handers. However, conclusions based on threshold-dependant group averages may be misleading as they ignore inter-individual differences.

### One-handers, but not amputees, show lip remapping in the deprived cortex based on univariate topographic mapping

To account for inter-individual differences in functional topography and brain topology, we calculated for each participant a winner-takes-all map across facial parts within a (combined) hand and face S1 region of interest (ROI). Focusing on the centre of gravity (CoG) of the lips cluster, we first explored changes in the cortical (geodesic) distance between the lips and an anatomical landmark (~1 cm lateral to the hand knob) of amputees and controls. Here we found no statistically significant main effects or group x hemisphere interaction ($F_{(1,36)}$=0.019, p=0.890, $n^2_p$=0.001, $BF_{10}$=0.297; controlled for brain size volume; *Figure 2B*), indicating that the lip area in amputees is not located differently to that of controls. We also measured the proportion of the deprived hand ROI occupied by the lip-winner surface area (relative to the intact hand area; Laterality index). We did not find a significant remapping of the lips (i.e., greater surface coverage) in the missing hand ROI for amputees when compared to controls ($U$=141.000, p=0.197, $r_b$ = 0.246, $BF_{10}$=0.579; *Figure 2C*) or to zero ($W$=91.000, p=0.245, $r_b$=0.338, $BF_{10}$=0.472; *Figure 2C*), though here the Bayes Factors did not provide conclusive evidence. Together, these results suggest, contrary to popular theories on brain plasticity in amputees (*Flor et al., 2006*), that the lips do not remap into the deprived hand area. We next compared the lips laterality index between those individuals who reported suffering from PLP (n=11) and those who no longer experienced chronic PLP (n=6) and found no significant differences ($U$=27.000, p=0.295, $r_b$ = 0.182, $BF_{10}$=0.629).

When visualising the average lip activity within the one-handers group, however, we did note a slight qualitative shift in the location, and spread, of the lip activity within the deprived hemisphere (*Figure 1A*). This is further supported by a visible shift of the one-handers lip-winner consistency map towards the deprived hand area (*Figure 2A*). These qualitative changes in the lip representation resulted in a significant group x hemisphere interaction for the lips cortical distance to the anatomical landmark in one-handers and controls ($F_{(1,40)}$=4.352, p=0.043, $n^2_p$=0.098; controlling for

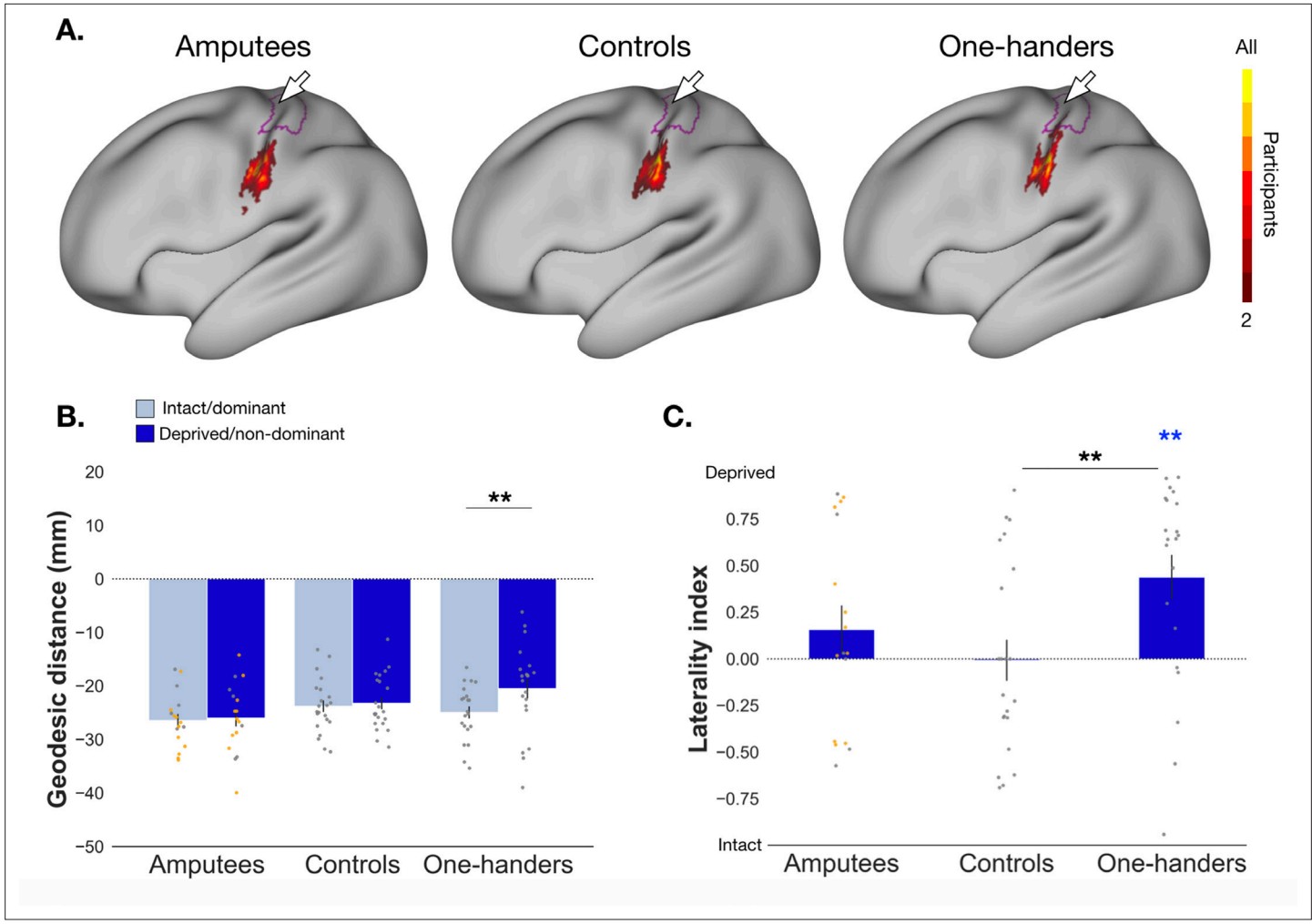

**Figure 2.** Characterisation of lip (re)mapping in the primary somatosensory cortex. (**A**) Group-level consistency map for the lips clusters resulting from the individual winner-takes-all maps in the S1 ROI (defined by combining the hand and face areas). The colour gradient represents participant agreement for the lips 'winning' that particular voxel, relative to other face movements. Please note that the individual-participant winner-takes-all maps are minimally thresholded, and thus produce an inherently different spatial distribution relative to the group contrast maps presented in *Figure 1*. The hand ROI is outlined in purple and central sulcus denoted by the white arrow. (**B**) Cortical geodesic distances from the lip CoG to the anatomical landmark (~1 cm lateral to the hand knob) are plotted for amputees (n=17), controls (n=22), and one-handers (n=21). Distances in the intact/dominant hemisphere are plotted in light blue, and distances in the deprived/non-dominant hemisphere are plotted in darker blue (in amputees and one-handers/controls, respectively). Positive distances indicate the lips CoG is located medial to the anatomical landmark, and negative distances indicate the lips CoG is located lateral to that landmark. The anatomical landmark itself equates to a geodesic distance of zero. For main effects of comparison between amputees and one-handers versus controls, see *Figure 2—source data 1–2*. (**C**) Laterality indices for the proportion of surface area coverage of the lips in the hand ROI for all groups (amputees, controls and one-handers). Positive values indicate greater surface area coverage in the deprived/non-dominant hemisphere relative to the intact/dominant hemisphere (in amputees and one-handers/controls, respectively), and negative values reflect greater surface area coverage in the intact/dominant hemisphere relative to the deprived/non-dominant hemisphere. Standard error bars and all individual data-points are plotted in grey and uncorrected for brain size. Amputees with PLP (yes/no) are plotted in orange. ** p<0.01; coloured asterisk's indicate values are significantly different from zero.

The online version of this article includes the following source data for figure 2:

**Source data 1.** Main effects and interaction for comparison of geodesic distances between amputees and controls for the lips.

**Source data 2.** Main effects and interaction for comparison of geodesic distances of the lips between one-handers and controls.

**Source data 3.** Raw data for cortical geodesic distances of the lips for amputees, controls, and one-handers.

**Source data 4.** Raw data for laterality indices of the lips for amputees, controls, and one-handers.

brain size; *Figure 2B*). Confirmatory comparisons indicated no statistically significant shifts of the lip CoG in the deprived hemisphere when compared to the controls non-dominant hemisphere ($t_{(40)}$=-1.178, p=0.246,~$d$ = −0.395, ~$BF_{10}$=0.148; corrected alpha = 0.025; uncorrected p-values reported; *Figure 2B*). Although when compared to their intact hemisphere, shorter distances from the lips to the hand area were found in the deprived hemisphere of one-handers ($t_{(40)}$=-3.374, p=0.002,~$d$ = −0.621), indicating evidence for lip remapping. These shifts in the deprived hemisphere were also reflected in significantly greater surface area coverage of the lips in the hand ROI when compared to controls ($t_{(41)}$==-2.762, p=0.009, d=−0.843; *Figure 2C*), which was significantly different from zero ($W$=1098.000, p=0.006, $r_b$ = 0.426). This converging evidence of lip remapping is in line with previous work in one-handers (*Hahamy et al., 2017*; *Amoruso et al., 2021*).

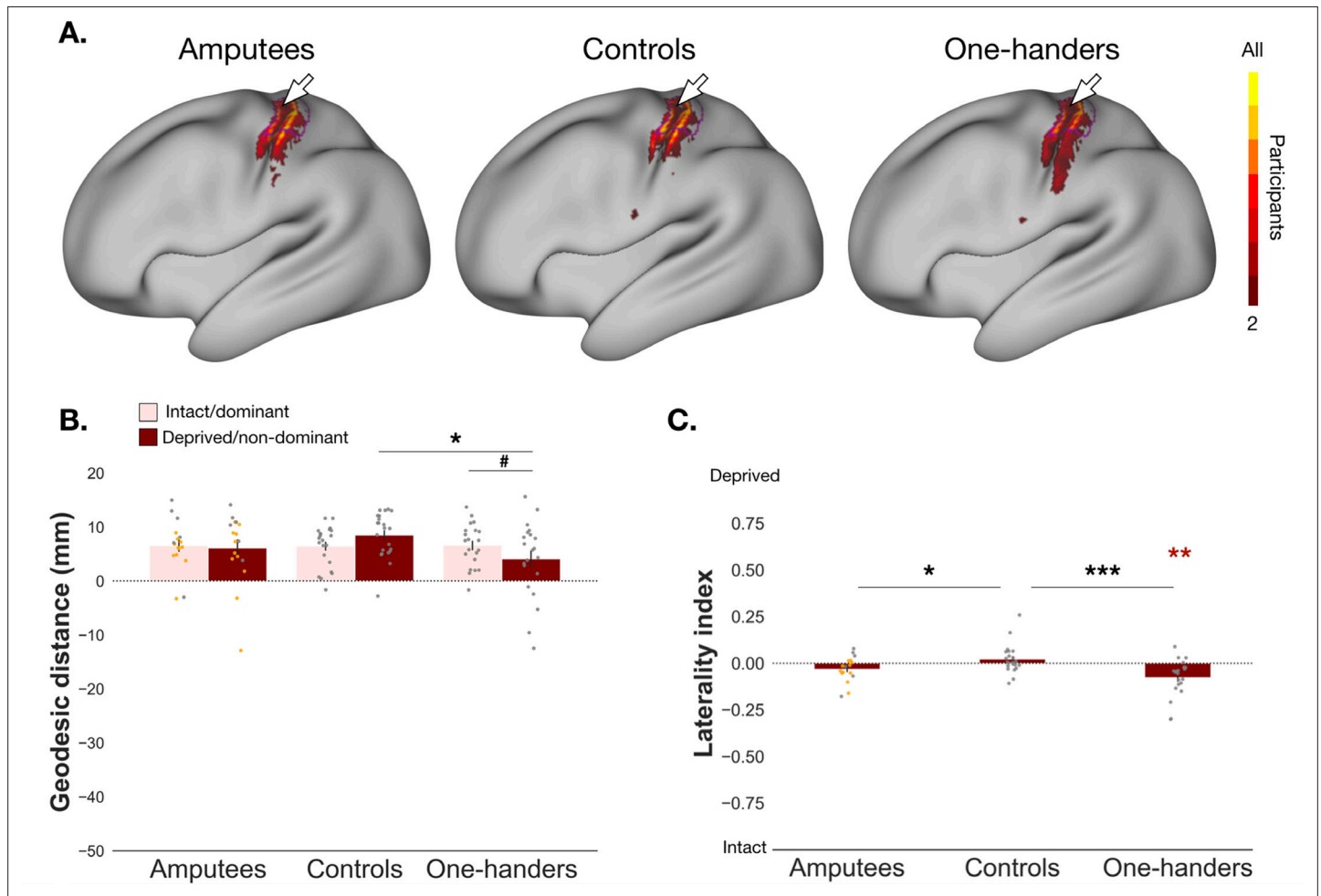

**Figure 3.** Characterisation of forehead (re)mapping in the primary somatosensory cortex. All annotations are as in *Figure 2*. For main effects of cortical geodesic distance comparison between amputees and one-handers versus controls, see *Figure 3—source data 1–2*. Distances in the intact/dominant hemisphere are plotted in pink and deprived/non-dominant hemisphere in red. (**B**) # p<0.05; * p<0.025 (corrected alpha); (**C**) * p<0.05; ** p<0.01; *** p<0.001; coloured asterisk's indicate values are significantly different from zero.

The online version of this article includes the following source data for figure 3:

**Source data 1.** Main effects and interaction for comparison of geodesic distances between amputees and controls for the forehead.

**Source data 2.** Main effects and interaction for comparison of geodesic distances between one-handers and controls for the forehead.

**Source data 3.** Raw data for cortical geodesic distances of the forehead for amputees, controls, and one-handers.

**Source data 4.** Raw data for laterality indices of the forehead for amputees, controls, and one-handers.

## One-handers, and to a lesser extent also amputees, show forehead remapping away from the hand area in the deprived cortex

As we note a qualitative upright orientation of the face (see *Figure 1*), the question remains as to whether the neighbour to the hand – the forehead – would reorganise after limb loss in amputees, as hypothesised by traditional theories (*Flor et al., 2006*). Again, we found no significant evidence for cortical remapping of the neighbouring forehead in amputees when assessing changes in cortical distances (group x hemisphere: $F_{(1,36)}$=1.338, p=0.255, $n^2_p$=0.036, $BF_{10}$=0.695; controlled for brain size volume; *Figure 3B*). But a significant difference was found for reduced forehead surface area coverage in the deprived hand ROI when compared to controls ($t_{(37)}$=2.236, $p$=0.031, $d$=0.722; *Figure 3C*). Interestingly, the direction of this effect indicates less, not more, remapping of the forehead in the deprived hand ROI of amputees. However, note that this decrease of surface area coverage was not significantly different from zero ($t_{(16)}$=-1.86, p=0.082, d=−0.451, $BF_{10}$=1.012; *Figure 3C*). When comparing the forehead laterality index for amputees with and without PLP, no significant differences were found ($t_{(15)}$=-0.729, p=0.761, d=−0.370, $BF_{10}$=0.291). Taken together, these results suggest that if remapping of the cortical neighbour – the forehead – does occur, this occurs away from the hand area, and is not related to PLP.

When looking at the one-handers group we did find significant evidence of forehead remapping with a group x hemisphere interaction ($F_{(1,40)}$=7.437, p=0.009, $n^2_p$=0.157; controlled for brain size volume; *Figure 3B*). Confirmatory comparisons indicated a positive trend for shorter distances of the foreheads' CoG to the anatomical landmark in the deprived hemisphere when compared to their intact hemisphere ($t_{(40)}$=2.085, p=0.043,~d = 0.435,~$BF_{10}$=1.18; corrected alpha = 0.025; trend defined as p<.05; uncorrected p-values reported) and significantly shorter distances when compared to the controls non-dominant hemisphere ($t_{(40)}$=2.580, p=0.014,~d = 0.774). As the forehead's CoG tended to be located above the anatomical landmark (see *Figure 3A*), these results indicate a significant shift of forehead activity away from the deprived hand ROI. This is further supported by a significant decrease of surface area coverage for the forehead in the deprived hand ROI when compared to controls ($t_{(41)}$=3.676, p<.001, d=1.122), which was significantly different from zero ($t_{(20)}$=-3.57, p=0.002, d=−0.779; *Figure 3C*). Remapping of the cortical neighbour in one-handers, therefore, manifests in a shifting away of the upper-face from the deprived hand area, possibly due to increases in activity of other facial movements, for example lips.

## Tongue movements produce different topographic maps across groups

We also assessed changes in the tongue representation, which is not an immediate neighbour to the hand in S1 (*Figure 4A*). We did find evidence for significant shifts in the tongue's CoG towards the anatomical landmark in amputees when compared to controls (group x hemisphere: $F_{(1,36)}$=4.859, p=0.034, $n^2_p$=0.119; controlled for brain size volume; *Figure 4B*). Confirmatory comparisons indicated significantly shorter distances in the deprived hemisphere of amputees when compared to their intact hemisphere ($t_{(36)}$=-2.595, p=0.014,~d = 0.678) but not to the controls non-dominant hemisphere ($t_{(36)}$=1.690, p=0.100,~d = 0.454,~$BF_{10}$=1.211; corrected alpha = 0.025; uncorrected p-values reported). The tongue showed only a trend for greater surface area coverage in the deprived hand ROI of amputees when compared to controls ($t_{(37)}$=-2.011, p=0.052, d=−0.650, $BF_{10}$=1.48; *Figure 4C*), and tended to be different to zero ($t_{(16)}$=-1.93, p=0.072, d=−0.467). As tongue remapping is not reflected consistently across analyses, and due to the lack of pre-existing hypotheses, this preliminary result should be interpreted with caution. However, it does indicate that some level of cortical remapping may occur in amputees after limb loss.

We next explored whether this trend for an increase in tongue activity within the deprived hand ROI, as captured by the laterality index of amputees, was related to PLP (*Figure 4C*), and found a non-significant difference (U=28.000, p=0.325, $r_b$ = 0.152, $BF_{10}$=0.589). These results suggest, along with an inconclusive Bayes Factor, that amputees with PLP may not report greater instances of tongue remapping, when compared to amputees without PLP.

When repeating the same analysis in one-handers, we also found a significant group x hemisphere interaction ($F_{(1,40)}$=8.536, p=0.006, $n^2_p$=0.176; controlled for brain size volume; *Figure 4B*) for the cortical distance between the tongue's CoG and the anatomical landmark. Confirmatory comparisons indicated significantly shorter distances to the anatomical landmark in the deprived hemisphere compared to the intact hemisphere ($t_{(40)}$=-3.794, p<.001, ~d = −0.677), as well as when compared to

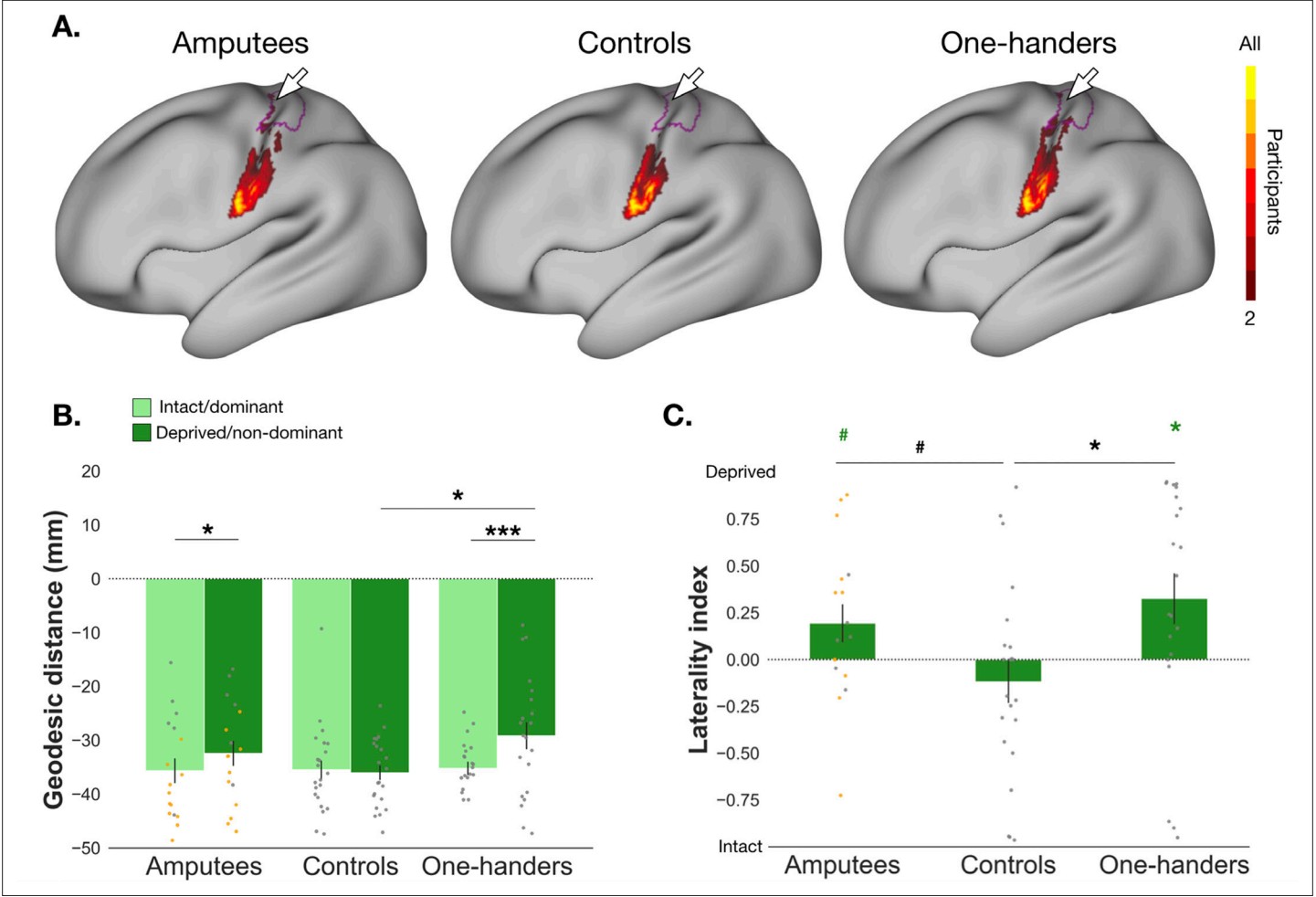

**Figure 4.** Characterisation of tongue (re)mapping in the primary somatosensory cortex. Distances in the intact hemisphere are plotted in light green and distances in the deprived hemisphere in dark green. For main effects of cortical geodesic distance comparison between amputees and one-handers versus controls, see *Figure 4—source data 1–2*. (**B**) * p<0.025 (corrected alpha); *** p<0.001; (**C**) # p<0.1; * p<0.05; coloured asterisk's indicate values are significantly different from zero. All other annotations are as in *Figure 2*.

The online version of this article includes the following source data for figure 4:

**Source data 1.** Main effects and interaction for comparison of geodesic distances between amputees and controls for the tongue.

**Source data 2.** Main effects and interaction for comparison of geodesic distances between one-handers and controls for the tongue.

**Source data 3.** Raw data for cortical geodesic distances of the tongue for amputees, controls, and one-handers.

**Source data 4.** Raw data for laterality indices of the tongue for amputees, controls, and one-handers.

the controls' non-dominant hemisphere ($t_{(40)}$=-2.380, p=0.022, ~$d = −0.751$). This was also reflected in greater surface area coverage of the tongue in the deprived hand ROI (see *Figure 4A*) that was significantly different from zero ($W$=162.000, p=0.035, $r_b = 0.543$) and from controls ($t_{(41)}$=-2.534, p=0.015, $d$=−0.773; *Figure 4C*). These results suggest that cortical remapping in one-handers extends to tongue movements.

### Nose movements produce similar topographic maps across groups

Similar analyses were performed to assess changes in the nose representation (see *Appendix 1—figure 1*). We did not find evidence for either CoG shifts (p values ≥ 0.829, $BF_{10}$ ≤0.337) nor differences in surface area coverage (p values ≥ 0.174, $BF_{10}$ ≤0.664) in both amputees and one-handers compared to controls (see *Appendix 1—figure 1*). These results suggest that the nose representation remains unaffected in both amputees and one-handers, with conclusive Bayes Factors for one-handers indicating evidence for the null.

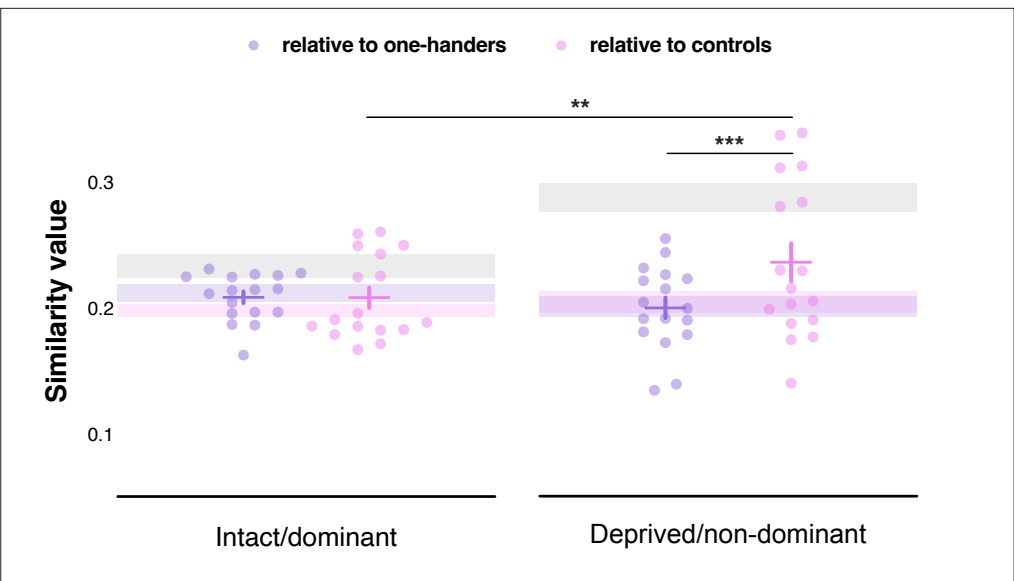

**Figure 5.** Jaccard similarity analysis comparing the winner-takes-all maps of amputees to the maps of controls and one-handers respectively. Similarity values indicate greater (towards 1) or reduced (towards 0) similarity between amputees' winner-takes-all maps (n=17) and those of controls (n=22; pink dots) or one-handers (n=21; purple dots), respectively. Results indicated significantly increased similarity to controls in the deprived hemisphere compared to one-handers' maps in the deprived hemisphere or to controls' maps in the intact hemisphere. For main effects and interaction, see *Figure 5—source data 1*. Means are plotted by the crosses, with standard error of the mean plotted along with individual data points. ** p<0.01, *** p<0.001. For reference, the intervals corresponding to the average intra-group similarities (+/-standard error) are represented by bands in the background (controls in grey, amputees in pink, and one-handers in purple). See *Figure 5—figure supplements 1–2* for follow-up Jaccard analyses.

The online version of this article includes the following source data and figure supplement(s) for figure 5:

**Source data 1.** Results from the linear mixed model comparing Jaccard similarity values of amputees' maps relative to the ones of one-handers and controls respectively.

**Source data 2.** Jaccard similarity values of amputees' maps relative to the ones of one-handers and controls, respectively.

**Figure supplement 1.** Jaccard similarity analysis comparing winner-takes-all maps both within- and across-groups.

**Figure supplement 2.** Jaccard similarity analysis comparing the winner-takes-all maps of controls to amputees and one-handers respectively.

## Amputees' topographic maps are more similar to the maps of controls than of one-handers

Finally, we wanted to investigate whether amputees' facial maps in the deprived hemisphere were more similar to those of controls or to one-handers. To provide a summary measure of univariate facial maps, we performed a Jaccard similarity analysis. This analysis quantifies the degree of similarity (0=no overlap between maps, 1=full overlap) between the map of each amputee and those of each individual in the controls or one-handers group respectively. A linear mixed model was used to compare the complete face map (including each of the facial sub-parts, see Methods). Results showed a significant group x hemisphere interaction ($F_{(1,240.0)}$=7.70, p=0.006; controlled for age; *Figure 5*), indicating that amputees' maps showed different similarity values to controls' and one-handers' depending on the hemisphere. Post-hoc comparisons (corrected alpha = 0.025; uncorrected p-values reported) revealed significantly higher similarity to controls' than to one-handers' maps in the deprived hemisphere ($t_{(240)}$=-3.892, p<.001). Amputees' maps also showed higher similarity to controls' maps in the deprived relative to the intact hemisphere ($t_{(240)}$=2.991, p=0.003). Amputees, therefore, displayed greater similarity of facial somatotopy in the deprived hemisphere to controls, suggesting again fewer evidence for cortical remapping in amputees.

However, it is important to note that the high intra-group similarity observed for controls (e.g. how similar controls' maps are to other controls' maps; grey bands in *Figure 5*) in the deprived hemisphere, could inflate the enhanced similarity to controls observed for amputees (see *Figure 5—figure supplement 1* to see all inter- and intra-group comparisons. Related to this we also observed that (i) one-handers also show an enhanced similarity to controls in the deprived hemisphere, and that (ii) even if lower than controls, amputees and one-handers show similar intra-group similarity in the deprived hemisphere). To account for this potential bias, we calculated the similarity between controls' maps (highly consistent across themselves) relative to amputees and one-handers respectively. Results showed a significant group x hemisphere x face-parts interaction ($F_{(3,315.0)}$=2.876, p=0.036; controlled for age; see *Figure 5—figure supplement 2*). Follow-up comparisons (corrected alpha = 0.006; uncorrected p-values reported) revealed that lip-winner maps of controls were significantly more similar to amputees' than to one-handers' lip maps in the deprived hemisphere ($t_{(315)}$=2.854, p=0.005). Conversely, controls' tongue-winner maps in the deprived hemisphere were significantly more similar to one-handers' maps than to amputees' maps ($t_{(315)}$=-2.883, p=0.004). Finally, controls' forehead-winner maps in the intact hemisphere were also significantly more similar to one-handers' maps than to amputees' maps ($t_{(315)}$=-3.576, p<.001). Altogether, these results confirm our previous quantification of univariate changes, with greater remapping of the lips in one-handers and if anything, remapping of the tongue in amputees.

## Brain decoding in the deprived hand area reveals stable facial representational pattern for amputees, and increased facial information in one-handers

The analyses described above focused on the topographic relationship of the four facial parts, but cortical remapping could potentially manifest subtly, without disrupting the spatial distribution of the face representation. RSA identifies statistical (dis)similarities across activity patterns, providing a more sensitive measure of representational changes (*Diedrichsen et al., 2018*).

When looking at face-face pairwise dissimilarity in the hand ROI across all three groups, we found a non-significant group x hemisphere x face-face interaction ($F_{(10,627.0)}$=0.572, p=0.837; controlled for age; *Figure 6*), suggesting a similar representational structure of the face across hemispheres and groups. However, when we looked at the average amount of facial information within the hand ROI, we did find a significant group x hemisphere interaction ($F_{(2,627.0)}$=14.544, p<.001), indicating potential differences in facial information content. Post-hoc comparisons (corrected alpha = 0.01; uncorrected p-values reported) exploring this effect reported significantly greater dissimilarity between facial-part representations in the deprived hemisphere of amputees (*M*=0.214; SE = 0.021; $t_{(627.0)}$=−4.401, p<.001) and one-handers (*M*=0.273; SE = 0.018; $t_{(627.0)}$=−5.668, p<.001), when compared to their respective intact hemisphere (amputees: *M*=0.152; SE = 0.0205; one-handers: *M*=0.202; SE = 0.018). When comparing to the controls' non-dominant hemisphere, we only found significantly greater facial information in the one-handers' deprived hand area ($t_{(72.8)}$=−3.297, p=0.002) and a non-significant effect for amputees ($t_{(71.7)}$=−0.828, p=0.411). We note that the effects observed in amputees may be influenced by reduced facial information in the intact hand (*M*=0.152, SE = 0.021; controls: *M*=0.207, SE = 0.017). While facial information in the deprived hand area was increased in one-handers compared with amputees, this effect did not survive our correction for multiple comparisons ($t_{(70.7)}$=−2.117, p=0.038). Similar, though weaker, results were obtained in M1 hand ROI (see *Figure 6—figure supplement 1* and *Figure 6—source data 2*). These results are in line with our univariate analyses, which demonstrate significant cortical remapping of facial parts in the one-handers group. In addition, these results indicate that there may be inter-hemispheric changes in facial information in the intact hand ROI of amputees, although this latter result awaits further confirmation.

For completeness, we also looked at facial activity patterns (i.e. face-face pairwise dissimilarities) within the face ROI across all three groups. Here we found non-significant differences for a group x hemisphere x face-face interaction ($F_{(10,627.0)}$=0.136, p=0.999) and group x hemisphere ($F_{(2,627.0)}$=0.626, p=0.535), suggesting a similar representational pattern of facial activity, that is facial information content, across hemispheres and groups (*Figure 7*). See *Figure 7—figure supplement 1* and *Figure 7—source data 2* for a similar analysis performed in M1 face ROI.

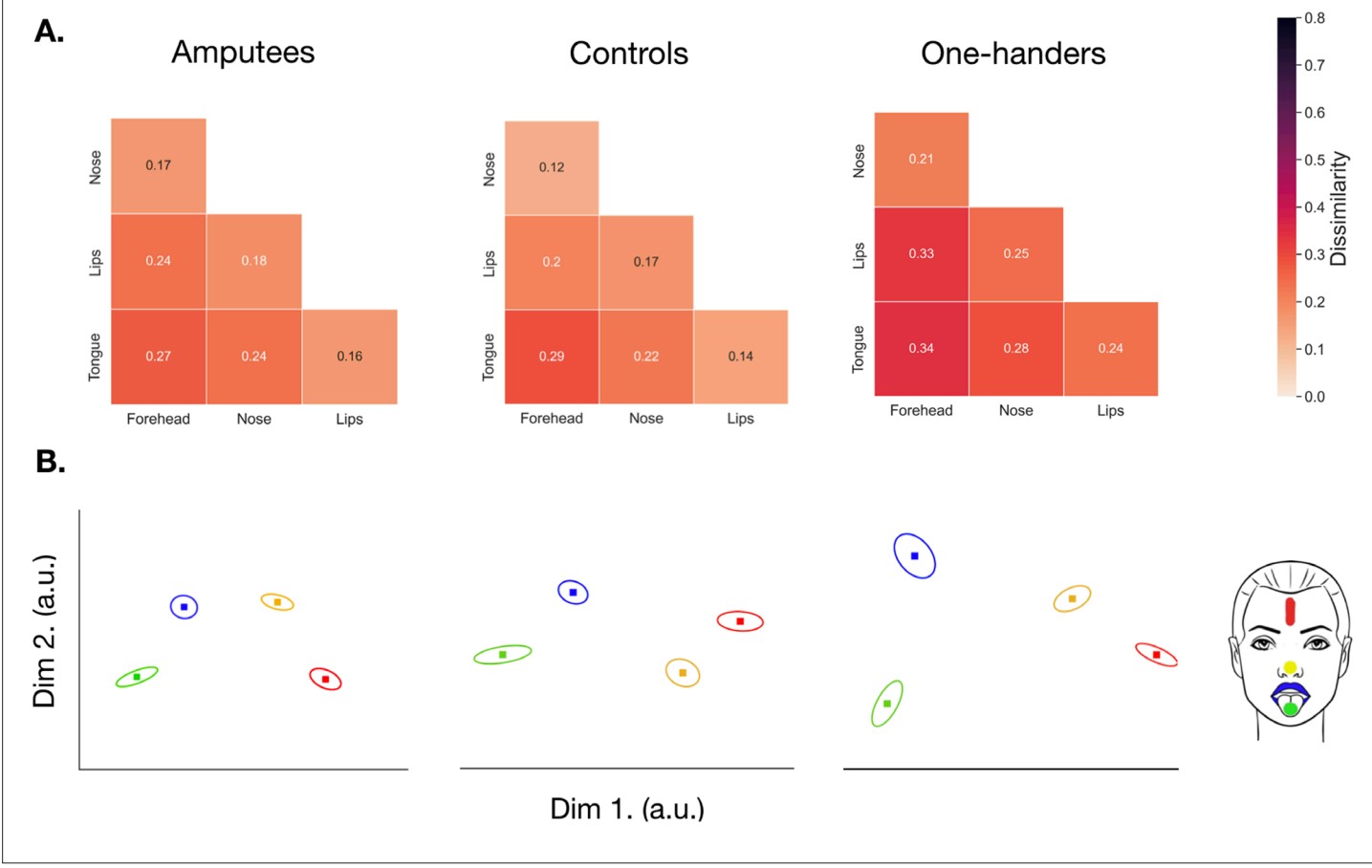

**Figure 6.** Representational Similarity Analysis (RSA) in the deprived/non-dominant hand area across all groups. (**A**) Representational Dissimilarity Matrices (RDMs) for amputees (n=17), controls (n=22), and one-handers (n=21). Greater dissimilarity between activity patterns for the chosen pairwise comparison indicates more information for that facial part within the hand area. Smaller dissimilarity values of facial activity patterns indicate a reduced ability to discriminate between the chosen movements in the hand area. (**B**) Multi-dimensional scaling plots for each group, which projects the RDM distances into a lower dimensional space. Here, the distances between each marker reflects the dissimilarity, with more similar activity patterns represented closer together, and more distinct activity patterns positioned further away. Forehead movements are plotted in red, with the nose in yellow, lips blue and tongue green, and the standard error is plotted around each data point. Please note, a different scale was used compared to the face ROI (*Figure 7*). For main effects and interaction for face-face pairwise distances in hand ROI see *Figure 6—source data 1*. For a similar analysis in M1 see *Figure 6—figure supplement 1* and *Figure 6—source data 2*.

The online version of this article includes the following source data and figure supplement(s) for figure 6:

**Source data 1.** Results from the linear mixed model used to explore differences in face-face pairwise distances in the hand ROI for amputees, one-handers, and controls.

**Source data 2.** Results from the linear mixed model used to explore differences in face-face pairwise distances in the M1 hand ROI for amputees, one-handers, and controls.

**Source data 3.** Multivariate distances for face-face pairs in the hand region-of-interest for amputees, one-handers, and controls.

**Source data 4.** Multivariate distances for face-face pairs in the hand region-of-interest for amputees, one-handers, and controls in M1.

**Figure supplement 1.** Representational Similarity Analysis (RSA) in the primary motor cortices deprived/non-dominant hand area across all groups.

## Discussion

It is a well-accepted notion, rooted in non-human primate electrophysiological data, that upper-limb amputation triggers cortical remapping of the assumed neighbour – the lower face – into the missing hand area. This previous work predominantly characterised remapping by investigating shifts of the lip representation (*Flor et al., 1995*; *Birbaumer et al., 1997*; *Lotze et al., 2001*; *Grüsser et al., 2001*; *Foell et al., 2014*; *Karl et al., 2001*; *MacIver et al., 2008*). However, by focusing on only one face part, activity elicited by other facial parts (such as the forehead) are not taken into account (see

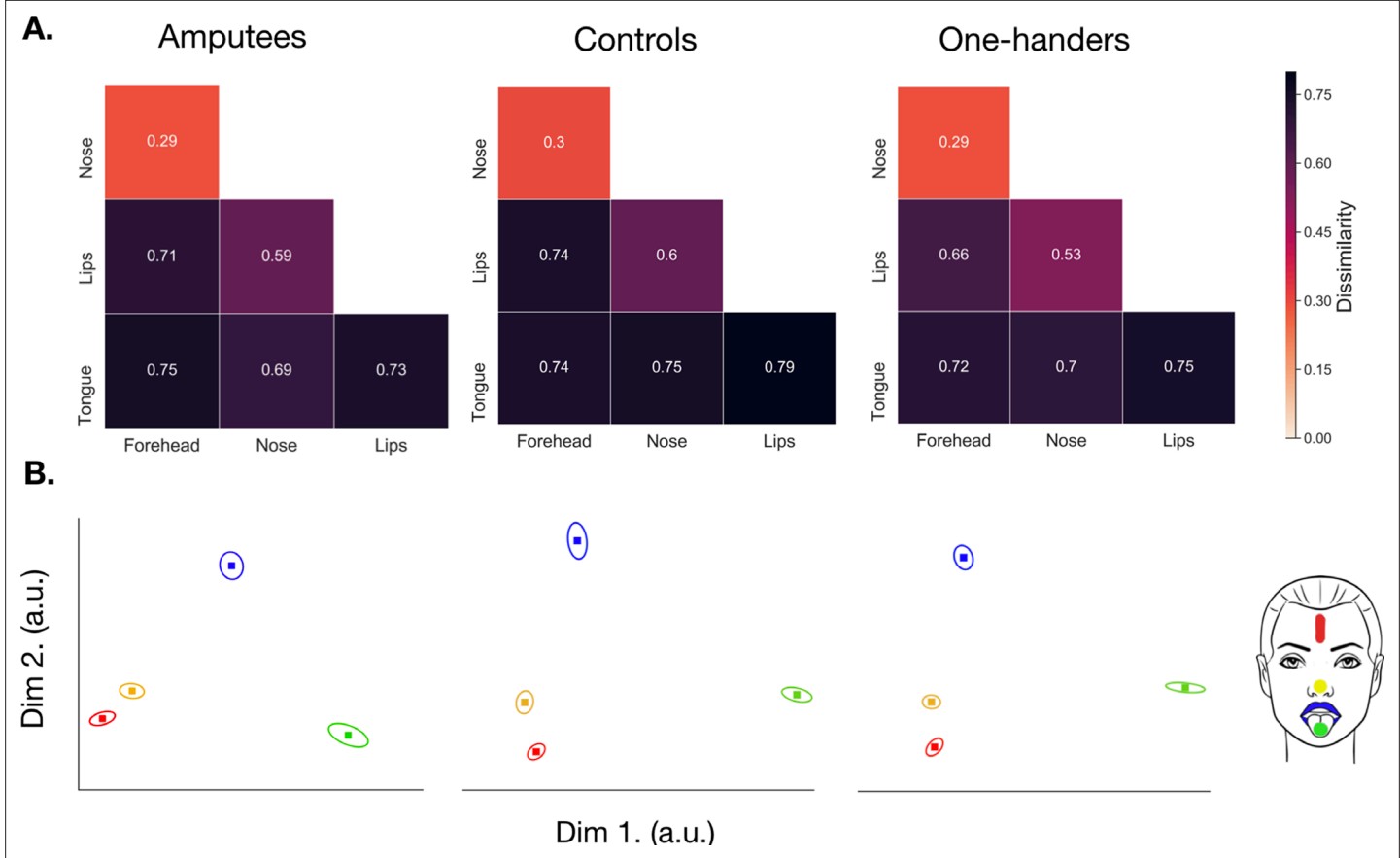

**Figure 7.** Representational Similarity Analysis (RSA) in the deprived/non-dominant face area across all groups. All annotations are as in *Figure 6*. For main effects and interaction for face-face pairwise distances in face ROI see *Figure 7—source data 1*. For a similar analysis in M1 see *Figure 7—figure supplement 1* and *Figure 7—source data 2*.

The online version of this article includes the following source data and figure supplement(s) for figure 7:

**Source data 1.** Results from the linear mixed model used to explore differences in face-face pairwise distances in the face ROI for amputees, one-handers, and controls.

**Source data 2.** Results from the linear mixed model used to explore differences in face-face pairwise distances in the M1 face ROI for amputees, one-handers, and controls.

**Source data 3.** Multivariate distances for face-face pairs in the face region-of-interest for amputees, one-handers, and controls.

**Source data 4.** Multivariate distances for face-face pairs in the face region-of-interest for amputees, one-handers and controls in M1.

**Figure supplement 1.** Representational Similarity Analysis (RSA) in the primary motor cortices deprived/non-dominant face area across all groups.

*Muret and Makin, 2021*). Here, we explored the relationship of face-to-hand remapping in controls and one-handed groups, and used both univariate (topographic) and multivariate (representational structure) methods to investigate in detail the information content of the face in both the deprived and intact hand and face areas. We found evidence for an upright somatotopy of the face across all groups, confirming that the cortical neighbour to the hand in humans is the upper, not lower, face. We further found little evidence for remapping of all tested facial parts in amputees, with no significant relationship to the presence of PLP. As a positive control, we also recruited individuals that were born without a hand (one-handers), who have previously shown cortical remapping across multiple body parts (*Hahamy et al., 2017*; *Hahamy and Makin, 2019*; *Amoruso et al., 2021*). Across multiple facial parts (forehead, lips and tongue), one-handers showed evidence for a complex pattern of face remapping in the deprived hand area, with consistent and converging evidence across analysis approaches. Finally, we demonstrate that facial representation in amputees' deprived hemisphere is more similar to two-handed controls than to one-handers. Together, our findings demonstrate that

the face representation in humans is highly plastic, but that this plasticity is restricted by the developmental stage of input deprivation, rather than cortical proximity.

Firstly, our univariate analyses at both group and individual levels provides converging and clear evidence for an upright orientation of the face in controls, amputees and one-handers. Contrary to previous neuroimaging studies reporting an inverted facial somatotopy (*Servos et al., 1999*; *Yang et al., 1993*; similar to primates *Manger et al., 1996*; *Jain et al., 2001*), or a lack of somatotopic organisation (*Iannetti et al., 2003*; *Nguyen et al., 2004*; *Kopietz et al., 2009*), here we found that the forehead representation borders the hand representation, followed by the nose, the lips, and the tongue located most laterally. These discrepancies may arise from the challenge to find a robust and reliable method to stimulate face parts, and thus elicit detectable cortical activation. In this context, it may be argued that it is difficult to achieve isolated execution of specific facial muscles when performing gross movements without impacting sensory processing of neighbouring facial parts. For instance, tongue movements in our paradigm (e.g. touching the roof of the mouth with the tongue), may be best considered as a holistic inner mouth movement, and forehead movements may be best considered as engaging the upper-face. While this critique is valid, it may also be relevant (although to a smaller degree) for passive paradigms, as stimulation can induce waves that propagate through the skin (*Manfredi et al., 2012*; *Sofia and Jones, 2013*; *Shao et al., 2016*) and Pacinian receptors were found to activate during stimulation of remote sites (*Edin and Abbs, 1991*; *Prsa et al., 2019*). Despite this caveat, both our univariate and multivariate analyses showed that we were successful in isolating sensorimotor representations of the various movements (forehead, nose, lips, and tongue) within our regions of interest (see *Appendix 1—figure 2* for validation of our approach using vibrotactile stimulation). In other words, even if somatosensory information is overlapping across movements, there is still enough distinct information to separate representational patterns. This finding indicates the suitability of our motor paradigm for teasing apart facial somatotopy, allowing us to characterise the face in greater detail than previously attempted. According to the confirmed upright organisation, if cortical remapping of neighbours exists, we would expect to see the forehead shifting towards and into the hand area – not the lips.

When looking at face-to-hand remapping in amputees, where the remapping of cortical neighbours has been the prevalent explanation for PLP, we find little evidence of shifts of locality and remapping towards the deprived hand area for facial parts, including the neighbour (forehead) and hypothesised neighbour (lips). This was further confirmed by the Jaccard analysis showing that amputees' maps were more similar to the ones of controls than of one-handers in the deprived hemisphere, as well as similar spatial representation of amputees' phantom thumb movements relative to controls (*Appendix 1—figure 3*). Our univariate results are further partially supported by our multivariate analysis, where we find no significant changes in dissimilarity of activity patterns across facial parts in amputees' deprived hand area relative to controls. These results support previous work reporting minimal cortical remapping after amputation (*Makin et al., 2013b*; *Kikkert et al., 2018*; *Raffin et al., 2012*), suggesting that in amputees this area might be functionally unipotent – pertaining to hand-related activity alone and lacking the ability to reorganise after hand loss. However, due to some inconclusive Bayes Factor in our key analyses, we cannot strongly conclude that remapping does not occur in this group. This could be attributed to our relatively small sample (further recruitment was prevented due to Covid-19 restrictions), and in particular, the small proportion of amputees experiencing PLP (11 out of 17). However, pain is not a necessary condition for deprivation-triggered remapping (*Recanzone et al., 1992*; *Wang et al., 1995*; *Andoh et al., 2020*) and vice versa, PLP has been shown to be experienced in absence of remapping (*De Nunzio et al., 2018*). Moreover, previous studies reporting significant difference between amputees who experienced PLP and those who do not, often employed similar (or smaller) sample sizes (*Lotze et al., 2001*), indicating that the expected effect of remapping should be substantial.

Our findings seem contradictory to the many previous studies reporting lip remapping in amputees (*Flor et al., 1995*; *Birbaumer et al., 1997*; *Lotze et al., 2001*; *Grüsser et al., 2001*; *Foell et al., 2014*; *Karl et al., 2001*; *MacIver et al., 2008*). A major difference with regard to these previous studies, which predominantly focused on a single part of the face, lies in the fact that our study was the first to assess the mapping and potential remapping of multiple facial parts at once. By focusing on the lips only, previous designs excluded other facial parts which may have elicited greater activity in certain areas of S1, resulting in a less accurate delineation of the lip-selective representation. Such

down-sampling of body maps, therefore, can lead to biased results and interpretation (*Muret and Makin, 2021*). While our design is not exempt from this limitation, the fact that we assessed other parts of the face may explain why our results diverged from previous findings.

We did find anecdotal evidence for remapping for the tongue within the deprived hand area in amputees. This was a surprising result, as the tongue is not a cortical neighbour to the hand and was not specifically hypothesised to remap in amputees. We also found that amputees demonstrated a different amount of facial information across the two hand areas. Although this multivariate result was not significantly different to that of controls, it demonstrates the plausibility that an inter-hemisphere imbalance may exist to a certain degree, albeit any relationship to PLP is tenuous. While these latter results require further validation, they support our premise that cortical proximity of representations may not be a necessity for remapping to occur. In this context, as our tongue condition could also be classed as an 'inner mouth' movement, it is important to note that previous work addressing sensorimotor representations of the mouth and the larynx have demonstrated both lateral and more medial 'hotspots' (i.e. a 'double' representation *Eichert et al., 2020*). The potential tongue remapping in amputees, therefore, may reflect changes in the medial mouth representation, but this would need to be investigated further. Moreover, even if the remapping we observed here goes against the theory of *cortical* proximity, it can still arise from representational proximity at the subcortical level, in particular at the brainstem level (*Florence and Kaas, 1995*; *Kambi et al., 2014*). While challenging in humans, mapping both the cuneate and trigeminal nuclei would be critical to provide a more complete picture regarding the role of proximity in remapping.

We did find converging and conclusive evidence for cortical remapping of multiple facial parts, both neighbours (forehead) and non-neighbours (lips and tongue) in our congenital one-handed group. Here, the pattern of remapping is strikingly different to that of cortical neighbourhood theories. Specifically, the location of the cortical neighbour – the forehead – is shown to shift away from the deprived hand area, which is subsequently more activated by the lips and the tongue than is the intact hand area. The increase of facial activity in the deprived hand area is in turn supported by our multivariate results, whereby significantly greater information content for the face was found in the deprived hand area for one-handers when compared to controls. One-handers' deprived hand area, therefore, seems to have increased discriminability between different facial movements. It is difficult to ascertain from our study the drivers of this remapping. It has been suggested previously that remapping within this group may be driven by functionally-relevant behaviour substituting for the loss of the limb (*Hahamy et al., 2017*; *Amoruso et al., 2021*). Together with the recent evidence that lip information content is already significant in the hand area of two-handed participants (*Muret et al., 2022*), compensatory behaviour since developmental stages might further uncover (and even potentiate) this underlying latent activity. Alternative explanations relate to an overall and unspecific release of inhibition (i.e. decreased GABA) in the missing hand area, allowing for latent activity of other body parts to be detected (*Hahamy et al., 2017*). While speculative, our results tend to support the former, as we report remapping for facial parts which have the ability to compensate for hand function, for example using the lips and/or mouth to manipulate an object, and a lack of remapping into the hand area for those that cannot (the forehead and nose). This increased activity from body parts compensating for hand function may represent a stabilising mechanism, aimed at preserving the integrity of the sensorimotor network and its function (*Muret and Makin, 2021*). The deprived hand area in one-handers, therefore, may reflect domain specificity – suitable for adapting to multiple body parts (*Amoruso et al., 2021*), which may preserve the role of the hand area by sustaining its hand-function related information content.

A limitation that should be acknowledged arises from the potential contribution to S1 from M1 activity. Since these cortical areas are neighbours, it is difficult to separate them with certainty. We minimised the contribution of M1 by taking multiple acquisition and pre-processing steps, including the use of anatomical delineation at the individual level, as well as a comprehensive analytical approach (e.g. both univariate and multivariate techniques). Perhaps most convincingly, our RSA evidence for remapping in congenital one-handers but not in amputees were qualitatively stronger in S1 relative to M1. Furthermore, it has been claimed that active movements may produce different cortical maps to those with passive stimulation (*Flor et al., 2013*; *Andoh et al., 2018*), and previous work demonstrating a relationship between cortical remapping and PLP tended to use passive stimulation (*Flor et al., 1995*; *Birbaumer et al., 1997*; *Foell et al., 2014*; *Karl et al., 2001*). However, we do not think

this methodological difference underlies our contrasting results as movement-induced lip activity has been shown to demonstrate lip remapping before (*Lotze et al., 2001*; *MacIver et al., 2008*; *Striem-Amit et al., 2018*), indicating that an active paradigm is suitable for assessing cortical remapping (if it exists). Conversely, a recent study using passive lip stimulation in amputees did not find any evidence for remapping (*Philip et al., 2017*). Moreover, we recently ran a study which found that S1 topography and multivariate representational structure are similar across active and passive paradigms (*Sanders et al., 2019*) (see *Appendix 1—figure 2* for a comparison between active and passive facial stimulation). In our view, the choice of an active paradigm is the most reflective of naturalistic tactile inputs in everyday life. Together with robust evidence for remapping in one-handers using all the methods tested here, our choice of active paradigm is clearly suitable to identify topographic organisation and remapping, and is practically accessible and translatable to fMRI designs.

To conclude, both our univariate and multivariate analyses found consistent evidence for a complex pattern of face remapping in congenital one-handers, in line with the theory suggesting remapping in this group reflects compensatory behaviour (*Muret and Makin, 2021*). This is in contrast to amputees, where we find little evidence for cortical remapping, indicating a relative stability of both the hand and face representation after limb loss. By and large, remapping measures were not linked to PLP. Our results call for a reassessment of traditional remapping theories based on cortical proximity, and future research into potential remapping of the inner mouth representation after limb loss.

## Materials and methods

### Participants

Seventeen individuals with acquired unilateral upper-limb amputation (age; $M$=53.71, SE = 2.69, women; n=4, missing right hand; n=9), twenty-one individuals with unilateral congenital transverse arrest (age; $M$=42.67, SE = 3.04, women; n=13, missing right hand; n=8) and twenty-two two-handed controls (age; $M$=45.55, SE = 2.02, women; n=10, left-handers; n=6) were recruited (see *Table 1* for full details). Two additional amputees who were recruited for the study did not participate in the scanning session due to MRI safety concerns, and further recruitment was stalled due to Covid-19 restrictions. The proportion of participants with intact/dominant right hand, as well as gender, were matched across groups ($\chi^2_{(2)}$=2.674, p=0.263; $\chi^2_{(2)}$=5.593, p=0.061). While significant differences between groups were observed for age ($H_{(2)}$=7.689, p=.021), post-hoc comparisons confirmed non-significant differences between amputees and one-handers relative to controls. Age covariates were therefore only included in statistical analyses when direct comparisons between amputees and one-handers were carried out. Procedures were in accordance with NHS National Research Ethics Service approval (18/LO/0474), and written informed consent was obtained.

### Phantom sensations rating

Amputees were asked to rate the frequency of PLP experience within the last year. They also rated the intensity of their worst PLP experience during the last week (or in a typical week involving PLP; 0=no pain, 100=worst pain imaginable). A chronic measure of PLP was calculated by dividing the worst PLP intensity in the last week by PLP frequency (1=all the time, 2=daily, 3=once a week, 4=several times per month, 5=once or less per month). This approach which takes into account the chronic aspect of PLP has been used successfully before (*Makin et al., 2015*; *Makin et al., 2013b*; *Kikkert et al., 2018*; *Draganski et al., 2006*; *Lyu et al., 2016*; *Kikkert et al., 2016*; *Kikkert et al., 2017*), and has high inter-session reliability (*Kikkert et al., 2018*). We also asked amputees about the vividness and frequency of non-painful phantom sensations (see *Table 1*).

### Functional MRI sensorimotor task

We used a facial active motor paradigm, where participants were visually instructed to move their forehead, nose, lips or tongue. This paradigm was chosen because it enabled bilateral activation of S1 simultaneously, allowing us to directly compare activity patterns between the two hemispheres (see *Appendix 1—figure 2* for validation of the active paradigm and Discussion for other considerations). Participants were also instructed to move their left and right thumb (amputees were asked to flex/extend their phantom thumb to the best of their ability; one-handers were asked to imagine

**Table 1.** Demographic details for amputees (A01-17) and congenital one-handers (C01-21).
Level of limb deficiency is as follows: 1=limb loss above elbow (transhumeral), 2=limb loss below elbow (transradial); L=left, R=right; PLS & PLP frequency: 0=no sensation or pain, 1=once or less per month, 2=several times per month, 3=once a week, 4=daily, 5=all the time. *PLP intensity rating was on average. PLS = phantom limb sensations; PLP = phantom limb pain.

| Participants | Age | Gender | Handedness (prior to amputation for amputees) | Affected limb | Level of limb deficiency | Years since amputation | PLS intensity | PLS frequency | Chronic PLS | PLP intensity | PLP frequency | Chronic PLP | Cause of amputation |
|---|---|---|---|---|---|---|---|---|---|---|---|---|---|
| AA01 | 60 | M | R | R | 2 | 43 | 100 | 5 | 100 | 60 | 5 | 60 | Trauma |
| AA02 | 34 | M | R | R | 1 | 3 | 50 | 2.5 | 14.6 | 70* | 2 | 17.5* | Trauma |
| AA03 | 58 | M | R | R | 1 | 33 | 90 | 5 | 90 | 100 | 1 | 20 | Trauma |
| AA04 | 59 | M | R | L | 2 | 16 | 40 | 1 | 8 | 0 | 1 | 0 | Trauma |
| AA05 | 54 | M | A | L | 1 | 36 | 100 | 5 | 100 | 80 | 4 | 40 | Trauma |
| AA06 | 47 | F | R | L | 2 | 18 | 80 | 4 | 40 | 0 | 0 | 0 | Electrocution |
| AA08 | 40 | F | R | R | 1 | 10 | 40 | 3 | 13.3 | 0 | 0 | 0 | Trauma |
| AA09 | 47 | M | R | R | 2 | 5 | 70 | 4 | 35 | 10 | 4 | 5 | Trauma |
| AA10 | 53 | M | R | L | 2 | 34 | 20 | 0 | 0 | 0 | 0 | 0 | Trauma |
| AA11 | 56 | F | L | L | 1 | 12 | 90 | 5 | 90 | 80 | 5 | 80 | Tumour |
| AA12 | 66 | M | R | R | 1 | 38 | 60 | 5 | 60 | 0 | 0 | 0 | Trauma |
| AA13 | 65 | F | L | L | 1 | 10 | 90 | 5 | 90 | 80 | 4 | 40 | Trauma |
| AA14 | 66 | M | R | L | 1 | 35 | 80 | 2 | 20 | 100 | 2 | 25 | Trauma |
| AA16 | 64 | M | R | R | 1 | 18 | 75 | 5 | 75 | 65 | 5 | 65 | Trauma |
| AA17 | 65 | M | R | R | 1 | 8 | 70 | 5 | 70 | 0 | 0 | 0 | Trauma |
| AA18 | 48 | M | R | R | 1 | 23 | 85 | 5 | 85 | 65 | 5 | 65 | Trauma |
| AA19 | 31 | M | R | L | 2 | 14 | 30 | 5 | 30 | 25 | 1 | 5 | Trauma |
| CA01 | 32 | F | R | L | 2 | | | | | | | | |

*Table 1 continued on next page*

*Table 1 continued*

| Participants | Age | Gender | Handedness (prior to amputation for amputees) | Affected limb | Level of limb deficiency | Years since amputation | PLS intensity | PLS frequency | Chronic PLS | PLP intensity | PLP frequency | Chronic PLP | Cause of amputation |
|---|---|---|---|---|---|---|---|---|---|---|---|---|---|
| CA02 | 32 | F | R | L | 2 | | | | | | | | |
| CA03 | 35 | M | R | L | 2 | | | | | | | | |
| CA04 | 48 | M | R | L | 2 | | | | | | | | |
| CA05 | 22 | F | L | R | 2 | | | | | | | | |
| CA06 | 54 | F | R | L | 2 | | | | | | | | |
| CA07 | 56 | F | L | R | 2 | | | | | | | | |
| CA08 | 53 | M | R | L | 1 | | | | | | | | |
| CA09 | 54 | F | R | L | 2 | | | | | | | | |
| CA10 | 58 | M | L | R | 2 | | | | | | | | |
| CA11 | 22 | M | R | L | 2 | | | | | | | | |
| CA12 | 30 | F | R | L | 2 | | | | | | | | |
| CA13 | 24 | M | L | R | 2 | | | | | | | | |
| CA14 | 33 | F | L | R | 2 | | | | | | | | |
| CA15 | 39 | F | L | R | 2 | | | | | | | | |
| CA16 | 55 | F | R | L | 2 | | | | | | | | |
| CA17 | 67 | F | L | R | 2 | | | | | | | | |
| CA18 | 30 | F | L | R | 2 | | | | | | | | |
| CA19 | 43 | M | R | L | 2 | | | | | | | | |
| CA20 | 63 | M | R | L | 2 | | | | | | | | |
| CA21 | 46 | F | R | L | 2 | | | | | | | | |

such movement), resulting in 6 conditions. Since the way one-handers performed the missing thumb condition could not be controlled (e.g., visual or motor imagery), this condition was not considered for analysis. Baseline (i.e. rest) was included as a 7th condition. Specific instructions involved: raising eyebrows (forehead), flaring nostrils (nose), puckering lips (lips), tapping tongue to the roof of the mouth (tongue), flexing and extending (thumb). The protocol comprised of 8 s blocks, with each condition repeated 4 times per run (5 times for baseline), over 3 functional runs. Before entering the scanner, participants practised each movement with the experimenter to ensure that the movement could be executed and to standardise each movement across participants (e.g. specificity and pace). Performance during scanning was visually monitored online with the use of an eye-tracker camera and an experimenter dedicated to this task. Note that multiple participants reported during the experimenter briefing that they could not successfully flare their nostrils, and were therefore instructed to attempt moving their nose in the scanner.

## MRI data acquisition

Functional and anatomical MRI data were obtained using a 3 Tesla Prisma MRI scanner (Siemens, Erlangen, Germany) with a 32-channel head coil. Anatomical data was acquired using a T1-weighted sequence (MPRAGE), with the following parameters: TR = 2530ms; TE = 3.34ms; flip angle = 7°; voxel size = 1 mm isotropic resolution. Functional data based on the blood oxygenation level dependant (BOLD) signal were acquired using a multiband T2*-weighted pulse sequence, with a between-slice acceleration factor of 4 and no in-slice acceleration (TR = 1450ms; TE = 35ms; flip angle = 70°; voxel size = 2 mm isotropic resolution; imaging matrix = 106 x 106; FOV = 212 mm). 72 slices were oriented in the transversal plane. A total of 172 whole-brain volumes for each of the three runs were collected per participant. Field-maps were acquired for field unwarping.

## MRI pre-processing

Functional data was first pre-processed using FSL-FEAT (version 6.00). Pre-processing included motion correction using MCFLIRT (*Jenkinson et al., 2002*), brain extraction using BET (*Smith, 2002*), temporal high-pass filtering with a cut-off of 119 s and spatial smoothing using a Gaussian kernel with a FWHM of 3 mm. Field maps were used for distortion correction of functional data.

For each participant, we calculated a midspace between the three functional runs, that is the average space in which the images are minimally reoriented. Each functional run was then aligned to the midspace and registered to structural images (within-subject) using FMRIB's Linear Image Registration Tool (FLIRT), and optimised using Boundary-Based Registration (*Greve and Fischl, 2009*). Where specified, functional and structural data were transformed to MNI152 space using FMRIB's Nonlinear Registration Tool (FNRIT *Andersson et al., 2010*).

## Functional MRI analysis

Time-series statistical analysis was carried out using FMRIB's Improved Linear Model (FILM). Task-based statistical parametric maps were computed by applying a voxel-based General Linear Model (GLM), as implemented in FEAT. The design was composed of 6 explanatory variables for each movement, convolved with a double-gamma hemodynamic response function (*Friston et al., 1998*), and its temporal derivative. The six motion parameters were included as regressors of no interest. Motion outliers (>0.9 mm) of large movements between volumes were included as additional regressors of no interest at the individual level (of total *n* volumes per group: amputees: 0.36%; controls: 0.36%; one-handers: 0.42%). For our main comparisons 6 contrasts were set up, corresponding to the facial movements' (forehead, nose, lips, tongue, and left/right thumb) relative to rest.

The estimates from the three functional runs were then averaged voxel-wise using a fixed effects model in participants structural space, with a cluster forming z-threshold of 2.3 and family-wise error corrected cluster significance threshold of p<0.05. Each estimates' average was masked prior to cluster formation with a sensorimotor mask, defined as the precentral and postcentral gyrus from the Harvard Cortical Atlas. The sensorimotor mask was registered to the individuals structural scan using an inversion of the nonlinear registration by FNIRT. All functional MRI analysis was carried in individual's native anatomical space.

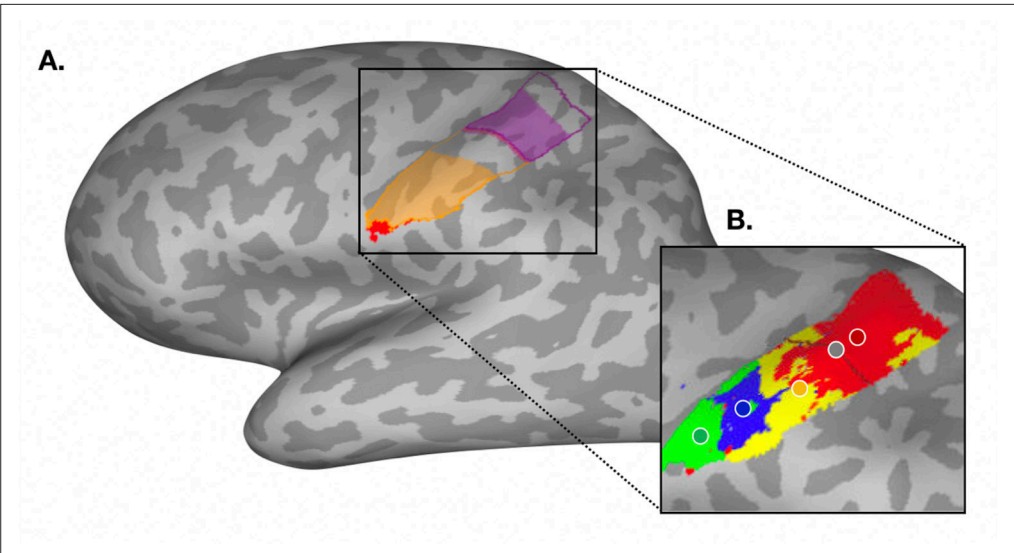

**Figure 8.** Regions of interest and winner-takes-all analysis in the primary somatosensory cortex for an example participant. (**A**) Regions of interest (ROIs) used for univariate analyses are outlined in purple for the hand, and in orange for the face. Shaded areas of each region of interest denote the trimmed ROIs used for multivariate analyses. ROI overlap with the secondary somatosensory cortex (**S2**) is highlighted in red, and was removed from the face ROI in order to minimise somatotopic contribution from that region. (**B**) A typical winner-takes-all map from an example participant, with forehead activity in red, nose activity in yelllow, lip activity in blue, and tongue activity in green. The centre-of-gravity for each movement is denoted by a coloured dot outlined in white. The anatomical landmark (used as an anchor for the CoG analysis) is outlined in black, with the midpoint denoted by a grey dot. Cortical geodesic distances were measured from each facial parts CoG to the anatomical landmark midpoint.

## Regions of Interest (ROI) definition

Facial topography and remapping were studied using anatomical ROIs for the hand and face areas in S1. Although the primary motor cortex (M1) and S1 are expected to activate during facial movement we primarily focused on S1 remapping due to the traditional focus in the maladaptive plasticity literature on S1 representational shifts (*Flor et al., 2006*). Furthermore, M1 topography tends to be less well-defined (*Schieber, 2001*; *Graziano and Aflalo, 2007*), and so characterisation of typical facial topography may be more apparent in S1. Nevertheless, we wish to note that due to the proximity of S1 to M1, it is possible that marginal contribution from M1 may have affected our S1 activity profiles.

Firstly, S1 was defined on the average surface using probabilistic cytoarchitectonic maps, by selecting nodes for Brodmann Areas (BAs) 1, 2, 3 a, and 3b (*Wiestler and Diedrichsen, 2013*). The S1 hand ROI (hereafter hand ROI) was defined by selecting the nodes approximately ~1 cm below and ~2.5 cm above the anatomical hand knob. In contrast to earlier work (*Wesselink and Ejaz, 2019*), we defined a conservative lateral boundary of the hand ROI (~1 cm below the hand knob) to ensure there was limited facial activity captured. From the remaining parts of S1, the medial region was discarded and the lateral region was selected as a first approximation of the S1 face ROI (hereafter face ROI; *Figure 8A*).

Structural T1-weighted images were then used to reconstruct pial and white-grey matter surfaces using Freesurfer (version 7.1.1) at the individual level. The hand and lateral ROIs were then projected into individual brains via the reconstructed individual anatomical surfaces. As the secondary somatosensory cortex (S2) contains a crude somatotopy (*Ruben et al., 2001*), the lateral ROI was further trimmed in participant's structural space by removing the overlap with S2. S2 was defined in MNI152 space using the Juelich Histological Atlas (*Wiech et al., 2014*). The S2 ROI was registered to participants' structural space using an inversion of the nonlinear registration carried about by FNIRT. The remaining lateral ROI with the overlap from S2 removed was used as the face ROI for all univariate analyses. We note that due to the probabilistic nature of these masks, there could be some marginal contribution from S2 in our estimated face area.

## Winner-takes-all approach

To characterise S1 facial topography, the hand and face ROIs were combined to produce an overall S1 ROI (minus the medial region), and a winner-takes-all approach was used (*Figure 8B*). For each participant, thresholded z-statistics averaged across the three functional runs were assigned to one of four face parts (forehead, nose, lips, tongue), dependent on which facial movement relatively showed maximal activity within the S1 ROI. Face-winners (i.e. the output of the winner-takes-all) were then projected to the individual's anatomical surface. Note that we excluded the thumb, which covered ~66% of the deprived hand ROI surface area in amputees and controls (see *Appendix 1— figure 3*). This allowed us to align our analysis with previous research, and to draw comparisons of facial somatotopy across all groups (one-handers do not have a phantom limb, and therefore we cannot probe the 'missing' hand representation directly). All subsequent analyses at the individual's anatomical surface level were computed using Connectome Workbench (v1.4.2).

## Cortical distance analysis

To assess possible shifts in facial representations towards the hand area, the centre-of-gravity (CoG) of each face-winner map was calculated in each hemisphere. The CoG was weighted by cluster size meaning that in the event of multiple clusters contributing to the calculation of a single CoG for a face-winner map, the voxels in the larger cluster are overweighted relative to those in the smaller clusters. The geodesic cortical distance between each movement's CoG and a predefined cortical anchor was computed. The cortical anchor was defined as the midpoint of the lateral border of the hand ROI (see *Figure 8B*). This anatomical landmark was drawn manually for each participant, the midpoint calculated, and both were visually confirmed by a second experimenter. The geodesic distance was assigned a negative value if the movement's CoG was located below the hand border (i.e. laterally).

## Surface area calculation

To assess possible remapping into the hand area, a secondary winner-takes-all analysis was restricted to the hand ROI only. The surface area coverage (mm$^2$) for each face-winner were computed on the individual anatomical inflated surface. We next calculated the proportion of the hand ROI occupied by each face part by dividing each face-winner's surface area by the total hand ROI surface area for each individual. From the resulting percentages, we produced a laterality index for each movement with the following formula:

$$laterality\ index = \frac{(deprived_m - intact_m)}{(\sum deprived_m \, , \, intact_m)} \tag{1}$$

whereby *deprived$_m$* and *intact$_m$* represent the percentage of surface area coverage for the facial movement *m*, respectively in the deprived and intact hemisphere. A subsequent laterality index of +1 indicates greater surface area coverage of that movement within the deprived hemisphere (or the hemisphere contralateral to the non-dominant hand in controls), whereas a value of 0 represents an equal balance of surface area coverage across both hemispheres. Note that this approach characterises cortical remapping in relation to the intact hemisphere and has been used in numerous previous studies on amputees (*Flor et al., 1995*; *Birbaumer et al., 1997*; *Grüsser et al., 2001*; *Foell et al., 2014*). It assumes that the intact hemisphere reflects baseline (i.e., that it is truly 'intact'), which may not be the case due to inter-hemisphere plasticity and/or homeostatic mechanisms (*Muret and Makin, 2021*; *Valyear et al., 2020*; *Philip and Frey, 2014*) and so also we compared our results to the control group.

## Jaccard analysis of similarity

To quantify the degree of similarity (0=no overlap between maps; 1=full overlap) between winner-takes-all maps across groups, we performed a Jaccard analysis of similarity between amputees' maps and those of controls and one-handers, respectively. For each face part winner map, the degree of similarity was calculated as follows (illustrated between amputees and controls):

$$Jaccard\ similarity = \frac{|amputee_{xm} \cap control_{ym}|}{|amputee_{xm} \cup control_{ym}|} \tag{2}$$

whereby *amputee*$_{xm}$ and *control*$_{ym}$ represent the winner-takes-all maps of the facial movement *m* of a given participant amputee *x* and control *y*. For each amputee, the similarity across the 22 controls and 21 one-handers respectively was averaged. The same approach was used to compare the other groups. For intra-group similarity, the participant analysed was excluded from the rest of its group to avoid comparing it to itself.

## Group-level visualisations

Prior to group-level visualisations, participant information regarding hand dominance (controls) and deprived hemisphere (one-handed participants) were used to sagittal-flip raw pre-processed data, such that the brain activity corresponding to the non-dominant/missing hand is always represented in the left hemisphere (note that a similar proportion of participants were flipped across groups). Group-level statistical parameter maps were then created with a threshold-free cluster enhancement (TFCE) approach using FSL's Randomise tool (*Winkler et al., 2014*). TFCE is a nonparametric, permutation-based method for cluster formation, and has been shown to demonstrate improved sensitivity when compared to typical thresholding methods (*Smith and Nichols, 2009*). Group activity mixed-effect maps were calculated for each fixed-effect (i.e. averaged across the three functional runs) parameter estimate of a face movement (forehead, nose, lips, tongue) contrasted to baseline. Prior to permutation (n=5000), parameter estimates were masked with a sensorimotor mask, defined as the precentral and postcentral gyrus from the Harvard Cortical Atlas. A family-wise error correction of p<0.05 and variance smoothing of 5 mm (as recommended for datasets with less than 20 participants) were used. Resulting clusters were thresholded at p<0.01 and projected to a group cortical surface (*Glasser et al., 2016*) using Connectome Workbench (v1.4.2), and activity is visualised in Brodmann Areas 1, 2, 3 a, 3b, and 4.

As well as activity maps, we also visualised the winner-takes-all output at the group-level. Here the 'winners' for each face movement within the S1 ROI (the hand and face region combined) in MNI152 space were concatenated into a single volume per group to produce a consistency map for the individual movements (i.e. how many participants maximally activated the same voxel when moving a given facial part). Resulting consistency maps were then projected to a group cortical surface (*Glasser et al., 2016*) using Connectome Workbench (v1.4.2) for visualisation only.

## Multivariate representational analysis

Representational Similarity Analysis (RSA *Nili et al., 2014*) was used to assess the multivariate relationship between activity patterns generated by each face part. RSA was conducted in the hand and face ROIs to explore possible remapping across representational features between groups. To ensure the selectivity of the hand and face areas, the ROIs used for univariate analyses were each further trimmed medially by ~1 cm, creating a 1 cm gap between the hand and face ROIs. For each participant, parameter estimates for the four facial movements (forehead, nose, lips and tongue) and the contralateral thumb (for controls and amputees only) were extracted from all voxels within the chosen ROI, as well as residuals from each runs' first-level analysis (three runs in total). Multidimensional noise normalisation was used to increase reliability of distance estimates (noisier voxels are down-weighted), based on the voxel's covariance matrix calculated from the GLM residuals. Dissimilarity between resulting facial activity patterns were then measured pairwise using cross-validated Mahalanobis distances (*Walther et al., 2016*). Due to cross-validation, the expected value of the distance is zero if two patterns are not statistically different from each other. Distances significantly different from zero indicate the two representational patterns are different; negative distances indicate noise. Larger distances for movement pairs therefore suggest greater discriminative ability for the chosen ROI. The resulting six unique inter-facial representational distances (10 unique distances when including the thumb for controls and amputees only) were characterised in a representation dissimilarity matrix (RDM). Multidimensional scaling (MDS) was also used to project the higher-dimensional RDM into lower-dimensional space, whilst preserving inter-facial dissimilarity, for visualisation purposes only. Analysis was conducted on an adapted version of the RSA Toolbox in MATLAB (*Nili et al., 2014*), customised for FSL (*Wesselink and Maimon-Mor, 2018*).

## Statistical analyses

All statistical analyses were carried out using JASP (Version 0.14). Outliers were classified as +/-3 standard deviations to the mean. We chose to not remove outliers in our analyses and checked that the significance and direction of the results did not change if identified outliers were removed (see below for number of outliers). When appropriate, univariate analyses was compared using parametric statistics. To assess normality for parametric tests, Shapiro-Wilk tests were run on residuals in combination with inspection of Q-Q plots and reporting of Levene's Test for Equality of Variances. Where stated, non-parametric test statistics are reported where the assumption of normality has been violated. Analysis of Variance (ANOVA) was used to explore group differences to controls in cortical distances (n=1 outlier for forehead in controls; n=1 outlier for tongue in controls). Each mixed ANOVA had a between-subject factor of Group (Controls x Amputees; Controls x One-handers) and a repeated-measures factor of Hemisphere (Intact/Dominant x Deprived/Non-dominant), and was run separately for each facial movement (forehead, nose, lips and tongue). We controlled for brain size volume when comparing cortical distances between groups. Post-hoc comparisons were conducted with a Bonferroni correction for multiple comparisons (reported corrected alpha and uncorrected p-values in text). Due to issues relating to integration of the correction for brain size in follow-up analyses, both the effect size (noted as $\sim d$) and Bayes factor (noted as $\sim BF_{10}$) in the post-hoc t-tests were not accounting for brain size. If assumptions of normality were violated, the difference between cortical distances in the intact and deprived hemisphere were calculated, and the group difference between one-handed groups and controls was computed using a Mann-Whitney U test. Resulting statistics are reported alongside the mixed ANOVA output. Independent *t*-tests were used to calculate laterality indices group differences (no outliers identified across groups). We reported the corresponding Bayes Factor ($BF_{10}$), defined as the relative support for the alternative hypothesis, for non-significant interactions and post-hoc comparisons. While it is generally agreed that it is difficult to establish a cut-off for what consists sufficient evidence, we used the threshold of $BF <1/3$ as sufficient evidence in support of the null, consistent with others in the field (*Wetzels et al., 2011*; *Dienes, 2014*) (though see *Kass and Raftery, 1995*). The Cauchy prior width was set at 0.707 (JASP's default). To investigate whether remapping measures were related to PLP, we used a one-tailed Mann-Whitney U tests to compare laterality indices of amputees with (n=11) and without PLP (n=6) for relevant facial parts, under the hypothesis that PLP should result in greater remapping (see *Appendix 1—figure 4* for the analogous analysis for the cortical geodesic distances).

For Jaccard analysis of similarity, a linear mixed model (LMM) analysis was used to compare the similarity of amputees' maps relative to those of controls and one-handers (no outliers), containing fixed factors of group (controls and one-handers), hemisphere (intact, deprived) and facial movement (forehead, nose, lips and tongue). A random effect of participant, as well as covariates of age, were also included in the model. A similar analysis was used to compare the similarity of controls' maps relative to those of amputees and one-handers respectively (n=1 outlier for the nose in controls). For multivariate analyses, the same analysis was used but with the fixed factors of group (controls, amputees, and one-handers), hemisphere (intact, deprived) and face-face pairs (6 unique representational distances; n=1 outlier in hand and face ROI for controls). All LMM's were carried out in Jamovi (version 1.6.15) under restricted maximum likelihood (REML) conditions with Satterthwaite adjustment for the degrees of freedom.

## Acknowledgements

We thank Arabella Bouzigues and Maria Kromm for their substantial help in terms of recruitment and data collection, we also thank Adriana Zainurin, Esther Teo, Christine Tan, Raffaele Tucciarelli and Mathew Kollamkulam for help with data collection. We thank Opcare for their help with participants recruitment, and our participants and their families for their ongoing support to our research. This work was supported by an ERC Starting Grant (715022 EmbodiedTech) and a Wellcome Trust Senior Research Fellowship (215575/Z/19/Z), awarded to TRM.

# Additional information

## Competing interests
Tamar R Makin: Senior editor, *eLife*. The other authors declare that no competing interests exist.

## Funding

| Funder | Grant reference number | Author |
|---|---|---|
| European Research Council | 715022 | Tamar R Makin |
| Wellcome Trust | 215575/Z/19/Z | Tamar R Makin |

The funders had no role in study design, data collection and interpretation, or the decision to submit the work for publication. For the purpose of Open Access, the authors have applied a CC BY public copyright license to any Author Accepted Manuscript version arising from this submission.

## Author contributions
Victoria Root, Conceptualization, Data curation, Software, Formal analysis, Validation, Investigation, Visualization, Methodology, Writing – original draft, Writing – review and editing; Dollyane Muret, Conceptualization, Data curation, Software, Formal analysis, Supervision, Validation, Investigation, Visualization, Methodology, Writing – original draft, Project administration, Writing – review and editing; Maite Arribas, Data curation, Formal analysis, Writing – original draft; Elena Amoruso, Investigation, Writing – original draft, Project administration; John Thornton, Aurelie Tarall-Jozwiak, Resources, Writing – review and editing; Irene Tracey, Supervision, Writing – review and editing; Tamar R Makin, Conceptualization, Resources, Supervision, Funding acquisition, Methodology, Writing – original draft, Project administration, Writing – review and editing

## Author ORCIDs
Victoria Root ⓘ http://orcid.org/0000-0002-0500-3206
Dollyane Muret ⓘ http://orcid.org/0000-0002-2626-654X
Tamar R Makin ⓘ http://orcid.org/0000-0002-5816-8979

## Ethics
Written Informed consent, and consent to publish, was obtained from all participants. Ethical approval was obtained from the NHS National Research Ethics Service approval (18/LO/0474).

## Decision letter and Author response
Decision letter https://doi.org/10.7554/eLife.76158.sa1
Author response https://doi.org/10.7554/eLife.76158.sa2

# Additional files

## Supplementary files
• Transparent reporting form

## Data availability
The data generated and analysed during this study is available to the public on Open Science Framework (https://osf.io/xq3am/).

The following dataset was generated:

| Author(s) | Year | Dataset title | Dataset URL | Database and Identifier |
|---|---|---|---|---|
| Root V, Muret D, Arribas M, Amoruso E, Thornton J, Tarall-Jozwiak A, Tracey I, Makin TR | 2021 | Complex pattern of facial remapping in somatosensory cortex following congenital but not acquired hand loss | https://osf.io/xq3am/ | Open Science Framework, 10.17605/OSF.IO/XQ3AM |

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

# Appendix 1

## Supplementary Results

### Nose movements produce similar topographic maps across groups

We also assessed changes in the nose representation (**Appendix 1—figure 1**). We did not find evidence for shifts in the nose's CoG towards the anatomical landmark in amputees when compared to controls (group x hemisphere: $F_{(1,36)}$=0.047, p=0.829, $n^2_p$=0.001, $BF_{10}$=0.337, controlled for brain size volume; **Appendix 1—figure 1B**). Similarly, no significant differences in surface area coverage were found in the deprived hand ROI of amputees when compared to controls ($t_{(37)}$=-1.385, p=0.174, d=−0.447, $BF_{10}$=0.664; **Appendix 1—figure 1C**), which however tended to be different from zero ($t_{(16)}$=-2.09, p=0.052, d=0.506).

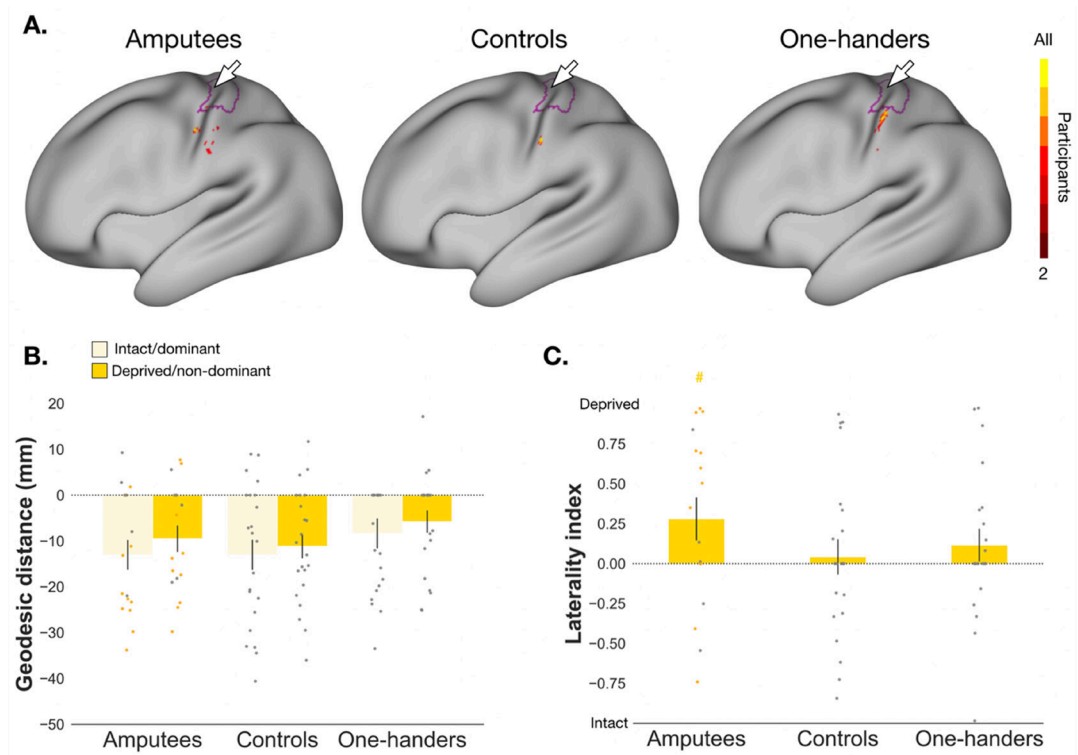

**Appendix 1—figure 1.** Nose remapping in amputees and one-handers in the primary somatosensory cortex. Distances in the intact hemisphere are plotted in light yellow and distances in the deprived hemisphere in dark yellow. All other annotations are as in **Figure 2**. # P<.1.

Similarly to amputees, we did not find any evidence for shifts in the nose's CoG towards the anatomical landmark in one-handers when compared to controls ($F_{(1,40)}$=0.033, p=0.857, $n^2_p$=0.001, $BF_{10}$=0.294; controlled for brain size volume; **Appendix 1—figure 1B**), nor differences in surface area coverage compared to controls (U=203.000, p=0.498, $r_b$ = 0.121, $BF_{10}$=0.332; **Appendix 1—figure 1C**). Taken together, these results suggest that the nose representation remains unaffected in both amputees and one-handers, with conclusive Bayes Factors for one-handers indicating evidence for the null.

### Passive validation of our active paradigm

In order to validate our active paradigm, two two-handed controls underwent the active paradigm twice, as well as a passive version of it involving a tactile stimulation (See Supplementary Methods). A significant positive relationship between the two active sessions was found for both participants (see **Appendix 1—figure 2B**), indicating a relatively stable representational pattern of facial activity in the face region across time. We also find a trend for a positive correlation between the second active session and the passive paradigm in one participant, that is significant in the second participant (see **Appendix 1—figure 2B**). This indicates a broadly similar representational structure of the face

when evoked using both movements and passive stimulation. Note, however, smaller dissimilarity values for the passive paradigm, indicating a reduction in facial information available overall (see *Appendix 1—figure 2A*).

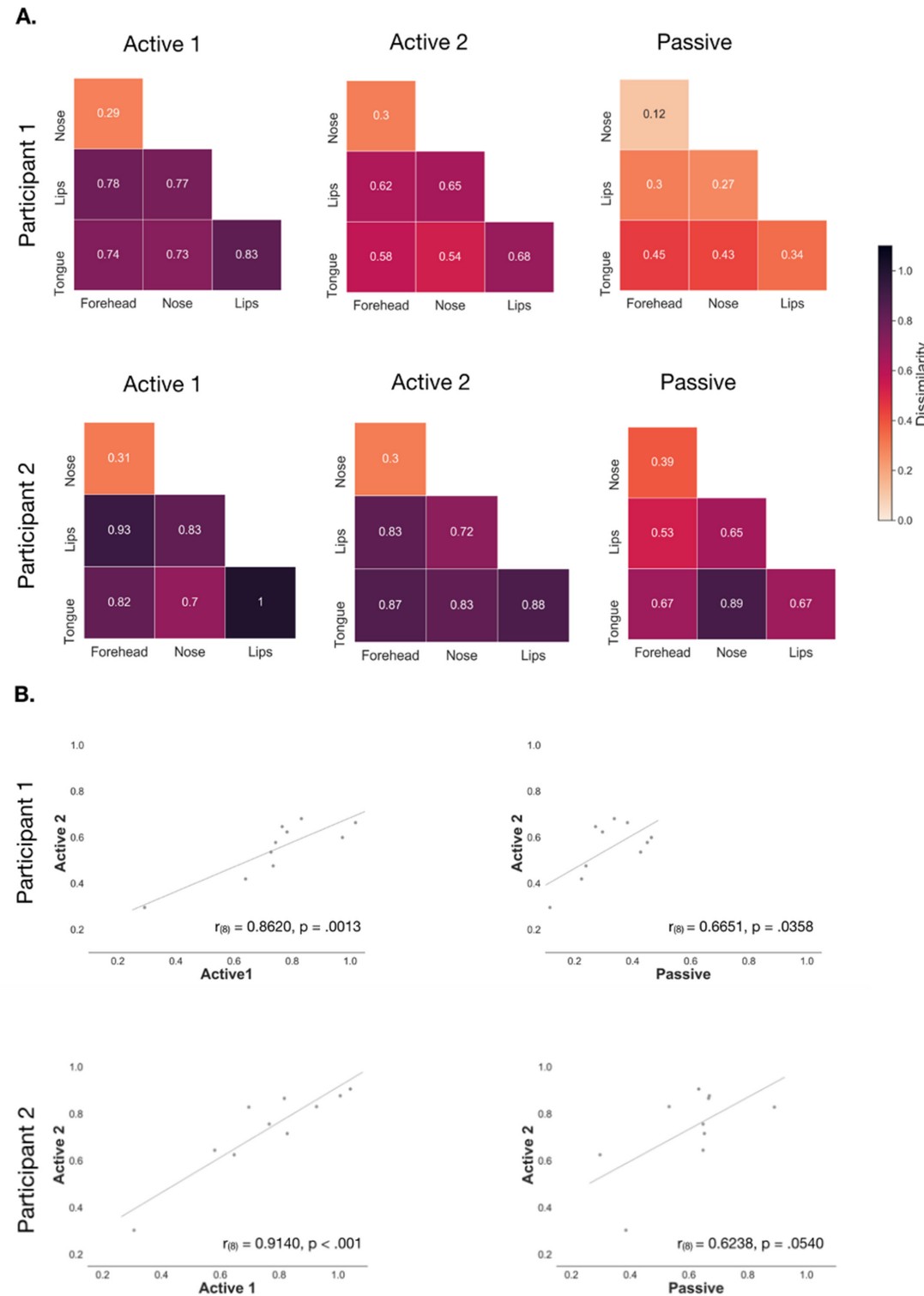

**Appendix 1—figure 2.** Comparison of active versus passive stimulation of the face for two control participants in the non-dominant hemisphere. (**A–B**) Representational Dissimilarity Matrices (RDMs) for two control participants in the face region of interest (see Methods; Multivariate representational analysis) who completed two active motor *Appendix 1—figure 2 continued on next page*

*Appendix 1—figure 2 continued*

paradigm (see Methods; Functional MRI sensorimotor task) sessions ~12 months apart and one passive session (see Supplementary Methods; Validation using passive stimulation). Greater dissimilarity between activity patterns for the pairwise comparison indicates an increased ability to discriminate between the two facial movements/ stimulations in the face region, that is there is a greater amount of facial information content. Smaller dissimilarity values indicated a reduced ability to discriminate between the two face parts. (C–D) Pearson's correlations examining the relationship between face-face and face-thumb dissimilarity values for both the first and second active motor paradigm sessions, and the second active session and passive paradigm.

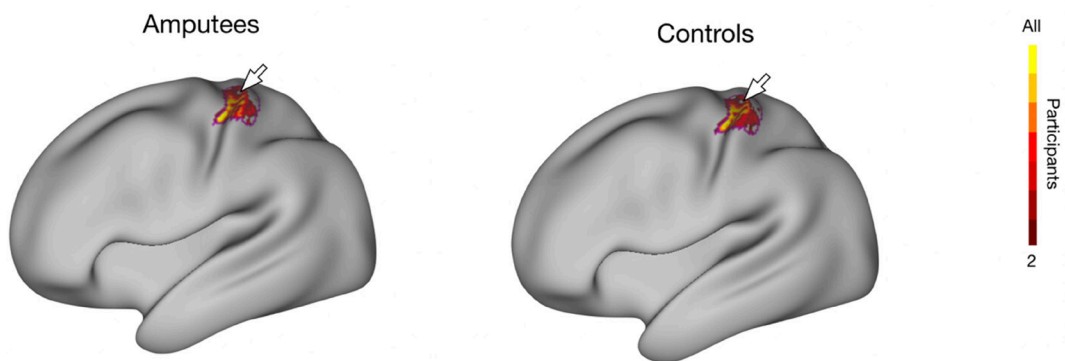

**Appendix 1—figure 3.** Phantom and non-dominant thumb representation in the deprived hemisphere of amputees and controls. Group-level consistency map for the phantom/non-dominant thumb in the hand ROI for amputees (n=17) and controls (n=22). Note the percentage of surface area coverage of the deprived/non-dominant hand ROI was not significantly different for the phantom (M=69.448%; SE = 3.752%) compared to the non-dominant thumb of controls (M=63.989%; SE = 3.594%; U=225.000, p=0.292, d=0.203, BF$_{10}$=0.579). The colour gradient represents participant agreement for maximally activating that particular voxel, relative to the face movements (winner-takes-all approach). The hand ROI is outlined in purple and central sulcus denoted by the white arrow.

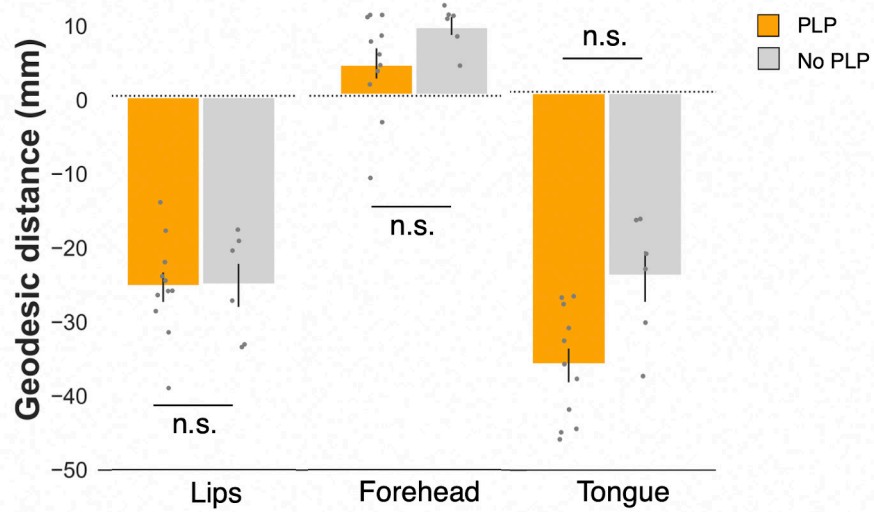

**Appendix 1—figure 4.** Comparison of cortical distances in the deprived hemisphere of amputees. Cortical (geodesic) distances were compared between amputees who reported the presence of PLP (n=11; orange) and amputees without PLP (n=6; grey) using a one-tailed Mann-Whitney test. Non-significant differences were found for the lips (t$_{(15)}$=-0.068, p=0.527, d=−0.035, BF$_{10}$=0.414), forehead (t$_{(15)}$=-1.720, p=0.947, d=−0.873, BF$_{10}$=0.203) and tongue (t$_{(15)}$=-3.018, p=0.996, d=−1.532, BF$_{10}$=0.156). Contrary to popular theories of brain plasticity and phantom limb pain (PLP; see Introduction), these results demonstrate that individuals with PLP do not exhibit greater instances of cortical remapping in the deprived hemisphere of the tested facial parts (including both the traditional marker of plasticity – the lips – and the cortical neighbour – the forehead).

## Supplementary Methods

### Validation using passive stimulation

#### Procedure

Two two-handed individuals took part in this validation procedure (aged 33 and 29, 2 women, 1 left-handed). In addition to the passive stimulation task described below (hereafter Passive), these individuals underwent two sessions of the sensorimotor active task used in the main analyses (hereafter Active1 and Active2). This allowed us to (i) assess the consistency of our data between the two active sessions and (ii) account for the fact that a different scanner was used for the passive task (and thus one of the active sessions, namely Active2, to be compared to the Passive session).

#### Functional MRI passive task

Soft pneumatic actuators (SPA-skin *Sonar et al., 2021*) were placed on the participants' forehead, nose, lips, tongue on the body midline, and on the thenar eminence of each thumb. Once the actuators had been attached and the participant was inside the scanner bore, a short thresholding procedure was carried out, whereby the pressure (KPa) of the actuators was adjusted to reach above-threshold subjective equality across body parts. The procedure began by defining the pressure for the least sensitive body part – the nose. Here pressure began at 35 KPa and was increased or decreased in step-sizes of 5 KPa. Stimulation length was 8 seconds and participants had 3 seconds to make their response, for example pressing a button to either increase/decrease the pressure or state that the pressure was 'ideal' (clearly perceived). The nose was then used as a reference body part, whereby 8 seconds of stimulation was administered to the nose followed by 8 seconds of stimulation to one of the other body parts (forehead, lips, tongue, left and right thumb), that is the target body part. The participant then had 3 seconds to respond via button-press in order to adjust the pressure (increasing or decreasing by 5 KPa) of the target body part in order to generate a similar tactile experience to that of the nose. Throughout thresholding, if the participant responded with 'ideal' twice, or if 5 trials had been administered, the resulting pressure was chosen for that body part. During the functional MRI task, each body part was stimulated using the thresholded KPa and a stimulation frequency of 5 Hz. The same block design (with similar time-course) as for the Active sessions was used (i.e., 8 s stimulation blocks, each condition repeated 4 times per run, 5 times for baseline), but over 4 functional runs instead of 3.

#### Functional MRI data acquisition and analysis

Functional and anatomical MRI data for the Passive and Active2 sessions were obtained with the same MRI parameters as for the Active1 session (collected on the same scanner as the data collected for the main analyses), but on a different 3T Prisma scanner. Functional data was pre-processed and analysed as described for the main analyses.

