## [Editor Report]

This fundamental work substantially advances our understanding of cortical remapping in people with congenital or acquired missing hands. The evidence supporting the idea that remapping may not follow cortical proximity but instead functional rules as to how the effector is used are compelling, with rigorous univariate and multivariate analyses applied to functional Magnetic Resonance Imaging data. Importantly, the authors suggest this is mostly the case for one-handers but not for amputees for who the reorganization seems more limited in general.

---

## [Decision Letter]

**Decision letter after peer review:**

Thank you for submitting your article "Reassessing face topography in primary somatosensory cortex and remapping following hand loss" for consideration by *eLife*. Your article has been reviewed by 3 peer reviewers, and the evaluation has been overseen by a Reviewing Editor and Christian Büchel as the Senior Editor. The following individuals involved in the review of your submission have agreed to reveal their identity: Moritz Wurm (Reviewer #1); Sliman J Bensmaia (Reviewer #2); Olivier Collignon (Reviewer #3).

Essential revisions:

After consultation with the other reviewers, we believe that this is an interesting and potentially important study but that significant clarifications are needed to fully convince us that the claims are indeed supported by the data. This opinion was reached because ultimately we feel additional analyses are required to address some of the key concerns raised.

All the points raised in the separate reviews should be addressed but here follows what are the key elements that were discussed in the consultation phase.

One concern links to the claim that it was unsolved whether the face representation in S1 is upside-down in humans as it has been shown in non-human primates. To us, it seems there is already ample evidence that an up-right organization of the face would be observed in the human S1. Even if the fMRI data in humans triggered some debates it seems it did not capture sufficient momentum to revisit the original upright face homunculus as described by Penfield itself, since most textbook representations continue to include upright face representations.

The originality of the current study compared to previous work done by the authors themselves should be clarified.

Given that a motor task was used it seems relevant to not only limit analyses to S1 but also include M1; the claim the M1 maps are less clear than those in S1 is not fully convincing.

The authors should provide additional data to support some of the claims made in the paper. In particular, the authors should be coherent in how they treat the nose condition across uni and multivariate analyses. Our feeling is that the nose should be reintroduced in the analytical pipeline and distance/overlap (both uni and multi-variate) should be calculated including the nose. If the authors decide to remove this condition, they should provide empirical support for doing so, and do it for all analytical steps if the reason is that the activity triggered by this condition was unreliable. Similarly, the reason to not include the thumb condition in some analyses and sometimes selectively in one-handers is unclear. This seems actually an interesting condition to fully explore, we guess for the same reasons the authors included the condition in the design. It seems the rationale relates to the fact one-handers have no fantom hand sensation but this was known beforehand, so if this is the reason, why include this condition?

The difference between the winner-takes-all analysis and the group consistency maps should be clarified.

Finally, there are concerns about the claim that one-handers differ from amputees when no direct statistical data support such a claim.

*Reviewer #1 (Recommendations for the authors):*

Using fMRI-based univariate and multivariate analyses, Root, Muret, et al. investigated the topography of face representation in the somatosensory cortex of typically developed two-handed individuals and individuals with a congenital and acquired missing hand. They provide clear evidence for an upright face topography in the somatosensory cortex in all three groups. Moreover, they find that one-handers, but not amputees, show shorter distances from lip representations to the hand area, suggesting a remapping of the lips. They also find a shift away of the upper face from the deprived hand area in one-handers, and significantly greater dissimilarity between face part representations in amputees and one-handers. The authors argue that this pattern of remapping is different to that of cortical neighborhood theories and points toward a remapping of face parts which have the ability to compensate for hand function, e.g., using the lips/mouth to manipulate an object.

These findings provide interesting insights into the topographic organization of face parts and the principles of cortical (re)organization. The authors use several analytical approaches, including distance measures between hand- and face-part-responsive regions and representational similarity analysis (RSA). Particularly commendable is the rigorous statistical analysis, such as the use of Bayesian comparisons, and careful interpretation of absent group differences.

There are only a few aspects that could be clarified or added:

Only in the discussion does it become clear that an active execution paradigm is used to map the somatotopy of facial parts. This could be clarified earlier, for example at the beginning of the Results section. The use of an execution paradigm seems to be a reasonable choice, which is supported by the robust topographic organization they find and the comparison between active and passive paradigms. However, the topographic organization is not restricted to S1; a similar organization can be found in the adjacent motor cortex. This could be mentioned in the Results, and perhaps a few words on these findings and/or the relationship between somatotopy and mototopy with regard to remapping might be informative.

It is a bit unclear why the nose is not included in the main winner-takes-all analysis (Figure 1) and only shown in the Supplementary Methods. The authors argue in the Methods section that they excluded the nose in the univariate analysis due to weak activation, whereas they include it in the RSA due to the increased sensitivity to subtle changes in activity patterns. This appears a bit inconsistent because also the RSA should suffer from weak voxelwise activations. In Supplementary Figure 1, the activation does not look particularly weak, and the winner-takes-all approach indicates that there are voxels that are most strongly activated for the nose compared to all other facial parts. This could be explained in more detail. Related to this point, did the authors test for nose remapping in amputees and one-handers?

What happened with the thumb movement imagery in the one-handers? Suppl. Figure 2 shows the respective effects for amputees and controls. Did the one-handers fail to show significant effects? It might be interesting to compare the one-handers' imagery effects with the execution effects in amputees and controls, even if putative differences might be less straightforward to interpret (might be due to representational differences or different paradigms).

Related to this point, the face-thumb distance values in Figure 5 are only shown for the controls and amputees. For the reader, the reason is not immediately clear, so perhaps this could be clarified in the figure caption.

There are differences between the forehead clusters found in the winner-takes-all analysis (Figure 1) and the group consistency maps; the latter showing a strong overlap with the hand ROI. For the lips and tongue, the maps appear much more consistent. What could be the reason for this discrepancy, and how can the overlap between hand ROI and forehead be interpreted?

When reporting the weak but significant result of reduced forehead coverage in amputees, the authors could also refer to the BF=1.5, which suggests only anecdotal evidence.

The authors did not remove outliers, which is justified, and the authors checked that the significance and direction of the results did not change if identified outliers were removed. However, could the authors indicate the number of outliers?

*Reviewer #2 (Recommendations for the authors):*

After amputation, the deafferented limb representation in the somatosensory cortex is activated by stimulation of other body parts. A common belief is that the lower face, including the lips, preferentially "invades" deafferented cortex due to its proximity to cortex. In the present study, this hypothesis is tested by mapping the somatosensory cortex using fMRI as amputees, congenital one-handers, and controls moved their forehead, nose, lips or tongue. First, they found that, unlike its counterpart in monkeys, the representation of the face in the somatosensory cortex is right-side up, with the forehead most medial (and abutting the hand) and the lips most lateral. Second, there was little evidence of "reorganization" of the deafferented cortex in amputees, even when tested with movements across the entire face rather than only the lips. Third, congenital one-handers showed significant reorganization of deafferented cortex, characterized principally by the invasion of the lower face, in contrast to predictions from the hypothesis that proximity was the driving factor. Fourth, there was no relationship between phantom limb pain reports and reorganization.

As a non-expert in fMRI, I cannot evaluate the methodology. That being said, I am not convinced that the current consensus is that the representation of the face in humans is flipped compared to that of monkeys. Indeed, the overwhelming majority of somatosensory homunculi I have seen for humans has the face right side up. My sense is that the fMRI studies that found an inverted (monkey-like) face representation contradict the consensus. Similarly, it is not clear to me how the observations (1) of limited reorganization in amputees, (2) of significant reorganization in congenital one-handers, and (3) of the lack of relationship between PLP and reorganization is novel given the previous work by this group. Perhaps the authors could more clearly articulate the novelty of these results compared to their previous findings. Finally, Jon Kaas and colleagues (notably Niraj Jain) have provided evidence in experiments with monkeys that much of the observed reorganization in the somatosensory cortex is inherited from plasticity in the brain stem. Jain did not find an increased propensity for axons to cross the septum between face and hand representations after (simulated) amputation. From this perspective, the relevant proximity would be that of the cuneate and trigeminal nuclei and it would be critical to map out the somatotopic organization of the trigeminal and cuneate nuclei to test hypotheses about the role of proximity in this remapping.

A few comments:

Which cortical fields do the authors refer to as primary somatosensory cortex? Brodmann's areas 3a, 3b, 1 and 2?

On line 322, I suggest using "completeness" rather than "completion."

On line 372, I suggest using "critique" rather than "limitation."

*Reviewer #3 (Recommendations for the authors):*

In their study, the authors set up to challenge the long-held claim that cortical remapping in the somatosensory cortex in hand deprived cortical territories follows somatotopic proximity (the hand region gets invaded by cortical neighbors) as classically assumed. In contrast to this claim, the authors suggest that remapping may not follow cortical proximity but instead functional rules as to how the effector is used. Their data indeed suggest that the deprived hand area is not invaded by the forefront which is the cortical neighbor but instead by the lips which may compensate for hand loss in manipulating objects. Interestingly the authors suggest this is mostly the case for one-handers but not in amputees for who the reorganization seems more limited in general (but see my comments below on this last point).

This is a remarkably ambitious study that has been skilfully executed on a strong number of participants in each group. The complementarity of state-of-the-art uni- and multi-variate analyses are in the service of the research question, and the paper is clearly written. The main contribution of this paper, relative to previous studies including those of the same group, resides in the mapping of multiple face parts all at once in the three groups.

In the winner takes all approach, the authors only include 3 face parts but exclude from the analyses the nose and the thumb. I am not fully convinced by the rationale for not including nose in univariate analyses – because it does not trigger reliable activity – while keeping it for representational similarity analyses. I think it would be better to include the nose in all analyses or demonstrate this condition is indeed "noisy" and then remove it from all the analyses. Indeed, if the activity triggered by nose movement is unreliable, it should also affect multivariate.

The rationale for not including the hand is maybe more convincing as it seems to induce activity in both controls and amputees but not in one-handers. First, it would be great to visualize this effect, at least as supplemental material to support the decision. Then, this brings the interesting possibility that enhanced invasion of hand territory by lips in one-handers might link to the possibility to observe hand-related activity in the presupposed hand region in this population. Maybe the authors may consider linking these.

The use of the geodesic distance between the center of gravity in the Winner Take All (WTA) maps between each movement and a predefined cortical anchor is clever. More details about how the Center Of Gravity (COG) was computed on spatially disparate regions might deserve more explanations, however. Moreover, imagine that for some reason the forefront region extends both dorsally and ventrally in a specific population (eg amputees), the COG would stay unaffected but the overlap between hand and forefront would increase. The analyses on the surface area within hand ROI for lips and forehead nicely complement the WTA analyses and suggest higher overlap for lips and lower overlap for forehead but none of the maps or graphs presented clearly show those results – maybe the authors could consider adding a figure clearly highlighting that there is indeed more lip activity IN the hand region.

In addition to overlap analyses between hand and other body parts, the authors may also want to consider doing some Jaccard similarity analyses between the maps of the 3 groups to support the idea that amputees are more alike controls than one-handers in their topographic activity, which again does not appear clear from the figures.

This brings to another concern I have related to the claim that the change in the cortical organization they observe is mostly observed in one-handers. It seems that most of this conclusion relies on the fact that some effects are observed in one-handers but not in amputees when compared to controls, however, no direct comparisons are done between amputees and one-handers so we may be in an erroneous inference about the interaction when this is actually not tested (Nieuwenhuis, 11). For instance, the shift away from the hand/face border of the forehead is also (mildly) significant in amputees (as observed more strongly in one-handers) so the conclusion (eg from the subtitle of the Results section) that it is specific to one-hander might not fully be supported by the data. Similar to the invasion of the hand territory from the lips which is significant in amputees in terms of surface area. All together this calls for toning down the idea that plasticity is restricted to congenital deprivation (eg last sentence of the abstract). Even if numerically stronger, if I am not wrong, there are no stats showing remapping is indeed stronger in one-handers than in amputees and actually, amputees show significant effects when compared to controls along the lines as those shown (even if more strongly) in one-handers. Also, maybe the authors could explore whether there is actually a link between the number of years without hand and the remapping effects.

One hypothesis generated by the data is that lips remap in the deprived hand area because lips serve compensatory functions. Actually, also in controls, lips and hands can be used to manipulate objects, in contrast to the forehead. One may thus wonder if the preferential presence of lips in the hand region is not latent even in controls as they both link in functions?

1) The authors should re-assess their conclusion that remapping is mostly present in one-handers than amputees in absence of a convincing statistical demonstration that it is actually the case. The amputees show similar effects or trends for similar remapping as those observed in one-handers. maybe one source of this lower reliability in the effects of amputees links to the lower number of participants and their higher variability in ethology (eg age of amputation etc…). Maybe the authors could explore whether there is actually a link between the number of years without hand and the remapping effects.

2) The authors should be coherent in the body part they include for their univariate and multivariate analyses. I let them decide whether they think the nose condition was too noisy (eg because people could not follow instructions to move the nose) or not. If they think the condition however did not elicit reliable activity, this should affect both uni and multivariate analyses.

3) I recommend the authors show the map elicited by hand movement in the three groups, support their claim that this is only reliably observed in controls and amputees but not one-handers; and maybe discuss (or even better do some analyses on that, eg correlation) the possibility that the preservation of the hand map is potentially a factor influencing remapping (eg do the amputees with the most salient remaining hand maps are those showing less remapping?).

4) The authors should clarify in their figure where do they observe that the lip increases activity WITHIN the hand region in one-handers.

5) The fact that the univariate results of Figure 1 do not follow well those presented in the WTA maps deserves some consideration. Indeed, what seems striking in Figure 1 is that the forehead activity extends more dorsally in one-handers, approaching the hand region, which contradicts conclusions from WTA. Also, it might be informative to have an outline of the brain mask (in all figures) of the S1 ROI used for constraining spatially the univariate analyses.

6) In Figure 5 the thumb is not represented in one-handers. It would be interesting for the reader to see that indeed the instruction to move the thumb did not elicit a reliable activity map and what was the criteria to selectively remove it in one-handers aside from the fact they don't experience fantom sensation. It is also unclear why the thumb is not displayed in the MDS while it's computed in the RDM of controls and amputees.

7) Maybe discuss further the potential (latent) link between hand and lips as they both can be used to manipulate objects even in controls.

---

## [Author Response]

Essential revisions:After consultation with the other reviewers, we believe that this is an interesting and potentially important study but that significant clarifications are needed to fully convince us that the claims are indeed supported by the data. This opinion was reached because ultimately we feel additional analyses are required to address some of the key concerns raised.All the points raised in the separate reviews should be addressed but here follows what are the key elements that were discussed in the consultation phase.One concern links to the claim that it was unsolved whether the face representation in S1 is upside-down in humans as it has been shown in non-human primates. To us, it seems there is already ample evidence that an up-right organization of the face would be observed in the human S1. Even if the fMRI data in humans triggered some debates it seems it did not capture sufficient momentum to revisit the original upright face homunculus as described by Penfield itself, since most textbook representations continue to include upright face representations.

We agree with the reviewers that the human’s neuroimaging data (not just using fMRI but also MEG) did not trigger an overwriting of the original findings from Penfield. But we think this reflects the disproportional impact of Penfield’s cartooning of the homunculus, despite the scarcity and inherent limitations of the data supporting his work and the (over-)simplified depiction of the Homunculus (as we detail below). We believe that the silent consensus is that sensorimotor body representation is more nuanced then revealed by this textbook cartoon, and as such, the idea that the hand area is neighbouring the lower face in humans have been strongly embedded in contemporary research in particular relating to brain reorganisation. As an illustrating example, the first paper to describe the lower face (lips and cheek) shifting towards the hand area (Flor et al., 1995), has been cited over 2,000 times. This is one of many related examples, and when put together there is no doubt that this misrepresentation is highly popular and has influenced not only development of treatment for phantom limb pain, but also other related condition, such as complex regional pain syndrome (Maihöfner et al., 2004, 2006), dystonia (Burman et al., 2009; Elbert et al., 1998) and spinal cord injury (Wrigley et al., 2009). This empirical ambiguity is due to the fact that the vast majority of human studies is focused on lip representation, combined with the methodological difficulties of stimulating the face in an MRI environment. Consequently, our study is among the first to replicate Penfield’s observations with converging neuroimaging analyses in individual participants.

Having said all that, to appease the reviewers, we have re-written the introduction to reflect the inclusive neuroimaging results, and nuanced our claim throughout the manuscript (starting with the title and abstract) to reflect our findings confirm the original observations made by Penfield and colleagues.

The originality of the current study compared to previous work done by the authors themselves should be clarified.

We are happy to clarify on the conceptual, methodological and empirical innovation our study provides. In short:

(1)Conceptually, it is crucial for us to understand if deprivation-triggered plasticity is constrained by the local neighbourhood, because this can give us clues regarding the mechanisms driving the remapping. We provide strong topographic evidence about the face orientation in controls, amputees and one-handers.(2)The vast majority of previous research on brain plasticity following hand loss (both congenital and acquired) in humans has exclusively focused on the lower face, and lips in particular. We provide systematic evidence for stable organisation and remapping of the neighbouring upper face, as well as the lower face. We also study topographic representation of the tongue (and nose) for the first time.(3)The vast majority of previous research on brain remapping following hand loss (both congenital and acquired, neuroimaging and electrophysiological) was focused on univariate activity measures, such as the spatial spread of units showing a similar feature preference, or the average activity level across individual units. We are going beyond remapping by using RSA, which allows us to ask not only if new information is available in the deprived cortex (as well as the native face area), but also whether this new information is structured consistently across individuals and groups. We show that representational content is enhanced in the deprived cortex one-handers whereas it is stable in amputees relative to controls (and to their intact hand region).(4)Based on previous studies, the assumption was that reorganisation in congenital one-handers was relatively unspecific, affecting all tested body parts. Here, we provide evidence for a more complex pattern of remapping, with the forehead representation seemingly moving out of the missing hand region (and the nose representation being tentatively similar to controls). That is, we show not just “invasion” but also a shift of the neighbour away from the hand area which has never been documented (or in fact suggested).(5)Using Bayesian analyses we provide definitive evidence against a relationship between PLP and forehead remapping, providing first and conclusive evidence against the remapping hypothesis, based on cortical neighbourhood.

Given that a motor task was used it seems relevant to not only limit analyses to S1 but also include M1; the claim the M1 maps are less clear than those in S1 is not fully convincing.

To explore M1 face representation, without having to make assumption about its underlying topography, we conducted Representational Similarity Analysis (RSA) in M1 (BA4) hand and face ROI’s (see resulting statistics below). RSA allows us to quantify and characterise functional organisation beyond the spatial attributes of selectivity maps. As detailed below and summarised in the new supplementary figures (Figure 6 and 7 —figure supplement 1), we find very similar representational structure for both face organisation and “reorganisation” across groups to what we reported originally in S1. The statistical evidence for remapping in the hand ROI were similar, though qualitatively stronger in S1 (i.e., significant difference between one-handers and two-handed controls only observed in S1).

The authors should provide additional data to support some of the claims made in the paper. In particular, the authors should be coherent in how they treat the nose condition across uni and multivariate analyses. Our feeling is that the nose should be reintroduced in the analytical pipeline and distance/overlap (both uni and multi-variate) should be calculated including the nose. If the authors decide to remove this condition, they should provide empirical support for doing so, and do it for all analytical steps if the reason is that the activity triggered by this condition was unreliable.

Following this comment, we re-ran all univariate analyses to include the nose, and updated throughout the main text and supplemental results and related figures. In short, adding the nose did not change the univariate results, apart from a now significant group x hemisphere interaction for the CoG of the tongue when comparing amputees and controls, better matching the trends for greater surface coverage in the deprived hand ROI of amputees. As such, we did not need to update our main interpretation based on this (extensive) reanalysis.

Similarly, the reason to not include the thumb condition in some analyses and sometimes selectively in one-handers is unclear. This seems actually an interesting condition to fully explore, we guess for the same reasons the authors included the condition in the design. It seems the rationale relates to the fact one-handers have no fantom hand sensation but this was known beforehand, so if this is the reason, why include this condition?

We did not intent the thumb condition in one-handers for analysis, as the task given to one-handers (imagine moving a body part you never had before) is inherently different to that given to the other groups (move – or at least attempt to move – your (phantom) hand). This condition was included solely to fill the experimental gap so that the protocols were matched as closely as possible across groups, as per our general practice (Makin et al., 2013 Nat. Comm.; Makin et al., 2013 *eLife*; Hahamy et al., 2017). As we note in our original manuscript, activity elicited by thumb movements dominates most of the hand area (66%) in both amputees and controls. To reduce the discrepancy in our analyses across all three groups, we cannot include this condition. For this reason, we decided to remove the hand-face dissimilarity analysis which we included in our original manuscript. Upon reflection we agreed that this specific analysis does not directly relate to the question of remapping (but rather of shared representation).

The difference between the winner-takes-all analysis and the group consistency maps should be clarified.

The individual-participant winner-takes-all maps are minimally thresholded, and thus produce an inherently different spatial distribution relative to the group contrast maps presented in Figure 1. Note that Figure 1 does not involve a winner-takes-all procedure, but rather random effect group contrasts (for each condition versus baseline separately). We made sure to clarify these differences.

Finally, there are concerns about the claim that one-handers differ from amputees when no direct statistical data support such a claim.

We are particularly grateful for this comment, we now provide clear evidence to demonstrate that amputees topographic face mapping is more similar to controls than to one-handers.

Reviewer #1 (Recommendations for the authors):Using fMRI-based univariate and multivariate analyses, Root, Muret, et al. investigated the topography of face representation in the somatosensory cortex of typically developed two-handed individuals and individuals with a congenital and acquired missing hand. They provide clear evidence for an upright face topography in the somatosensory cortex in all three groups. Moreover, they find that one-handers, but not amputees, show shorter distances from lip representations to the hand area, suggesting a remapping of the lips. They also find a shift away of the upper face from the deprived hand area in one-handers, and significantly greater dissimilarity between face part representations in amputees and one-handers. The authors argue that this pattern of remapping is different to that of cortical neighborhood theories and points toward a remapping of face parts which have the ability to compensate for hand function, e.g., using the lips/mouth to manipulate an object.These findings provide interesting insights into the topographic organization of face parts and the principles of cortical (re)organization. The authors use several analytical approaches, including distance measures between hand- and face-part-responsive regions and representational similarity analysis (RSA). Particularly commendable is the rigorous statistical analysis, such as the use of Bayesian comparisons, and careful interpretation of absent group differences.

We thank the reviewer for their positive and constructive feedback.

There are only a few aspects that could be clarified or added:Only in the discussion does it become clear that an active execution paradigm is used to map the somatotopy of facial parts. This could be clarified earlier, for example at the beginning of the Results section.

Thank you for pointing this out. We agree with the reviewer’s recommendation and have added the detail regarding active somatotopic mapping at the beginning of the Results section (line 154), as advised, and at the end of the Introduction (line 126).

The use of an execution paradigm seems to be a reasonable choice, which is supported by the robust topographic organization they find and the comparison between active and passive paradigms. However, the topographic organization is not restricted to S1; a similar organization can be found in the adjacent motor cortex. This could be mentioned in the Results, and perhaps a few words on these findings and/or the relationship between somatotopy and mototopy with regard to remapping might be informative.

We agree with the reviewer that the primary motor cortex (M1) is indeed relevant and informative for our results. To explore M1 face representation, we conducted Representational Similarity Analysis (RSA) in M1 (BA4) hand and face ROI’s (see resulting statistics below). RSA allows us to quantify and characterise functional organisation beyond the spatial attributes of selectivity maps. Since M1 is considered to be more broadly topographically organised relative to S1, we believe this approach is the most appropriate to compare the results across these two brain areas. As detailed below and summarised in the new supplementary figures (Figure 6 and 7 —figure supplement 1), we find very similar representational structure for both face organisation and “reorganisation” across groups to what we reported originally in S1. Interestingly, the statistical evidence for remapping in the hand ROI were similar, though qualitatively stronger in S1 (i.e., significant difference between one-handers and two-handed controls only observed in S1).

When looking at face-face pairwise dissimilarity in the M1 hand ROI (Figure 6 —figure supplement 1), we found a non-significant three-way interaction of Group (Amputees x Controls x One-handers) x Hemisphere (Intact x Deprived) x Face-Face pair (including the nose) (*F_(10,627.0)_*=0.398, *p*=0.948; controlled for age) and a significant Group x Hemisphere interaction (*F_(2,627.0)_*=7.553, *p*<.001). Post-hoc comparisons (corrected α=0.0125; uncorrected *p*-values reported) indicated a similar pattern of results to S1, with significantly greater dissimilarity between facial-part representations in the deprived hemisphere of amputees (*M*=0.214; *SE*=0.0193 ; *t_(627.0)_*=3.9633, *p*<.001) and one-handers (*M*=0.243; *SE*=0.0170; *t_(627.0)_*=3.8525, *p*<.001), when compared to their respective intact hemisphere (amputees: *M*=0.157; *SE*=0.0193; one-handers: *M*=0.193; *SE*=0.0170; corrected α=0.0125; uncorrected p-values reported). When comparing to controls non-dominant hemisphere (M=0.192; SE=0.0163), we did not find any significant differences in both one-handers (t_(77.4)_=2.1656, p=0.033) and amputees (t_(75.9)_=0.8371, p=0.405).These results are in line with both our multivariate and univariate results in S1, which demonstrate extensive cortical remapping of facial parts in our one-handers group, and tentatively suggests again inter-hemispheric changes in the intact hand ROI of amputees (amputees: *M*=0.157; *SE*=0.0193; controls: *M*=0.201; *SE*=0.0163; *t_(75.9)_*=-1.7382, *p*=0.086).

When looking at face-face pairwise dissimilarity in the M1 face ROI we note minor differences to our S1 multivariate results. Here, despite a non-significant Group x Hemisphere x Face-Face interaction (*F_(10,627.0)_*=0.2969, *p*=0.982; controlled for age; see Figure 7 —figure supplement 1), we found a significant Group x Hemisphere interaction (*F_(2,627.0)_*=4.3429, *p*=0.013), arising from lower facial information content in the intact (M=0.557; SE=0.0305) compared to the deprived (M=0.597; SE=0.0305) face ROI in amputees, but this difference did not survive our correction for multiple comparisons (t_(627.0)_=2.22199, p=0.027; corrected α=0.0125; trend defined as p<0.025; uncorrected p-values reported). Between-hemisphere differences were non-significant when looking at the one-hander group (deprived: *M*=0.536; *SE*=0.0268; intact: *M*=0.567; *SE*=0.0268; *t_(627.0)_*=-1.93153, *p*=0.054). Non-significant differences were also found when comparing the deprived face ROI in amputees (*t_(67.2)_*=-0.00525, *p*=0.996) and one-handers (*t_(68.0)_*=-1.64705, *p*=0.104) to the controls non-dominant hemisphere (*M*=0.597; *SE*=0.0257). These results therefore tentatively suggest that the reported interaction is likely driven by decreased facial information content in the intact face ROI when compared to the deprived face region in amputees (although note no significant differences were reported when comparing the intact face ROI in amputees to controls (amputees: *M*=0.557; *SE*=0.0305; controls: *M*=0.593; *SE*=0.0257; *t_(67.2)_*=-0.90806, *p*=0.367)), suggesting again the presence of inter-hemispheric plasticity within this group (akin to results in S1). We also surprisingly found a significant Group x Face-Face (*F_(10,627.0)_*=0.1934, *p*=0.038) interaction when looking at face-face pairwise distances in the M1 face ROI, suggesting that the information content for each movement (regardless of hemisphere) differed across groups. This interaction arose from significantly smaller forehead-lips distances for one-handers compared to controls (*t_(125)_*=-2.625, *p*=0.010), but this effect did not survive correction for multiple comparisons (corrected α=0.004).

These additional analyses are specified in lines 426-427, 442-444, in the legends of Figure 6 and Figure 7 of the main text, with their respective Figure supplements 1 and data source 2, and in the Discussion (line 574-576).

It is a bit unclear why the nose is not included in the main winner-takes-all analysis (Figure 1) and only shown in the Supplementary Methods. The authors argue in the Methods section that they excluded the nose in the univariate analysis due to weak activation, whereas they include it in the RSA due to the increased sensitivity to subtle changes in activity patterns. This appears a bit inconsistent because also the RSA should suffer from weak voxelwise activations. In Supplementary Figure 1, the activation does not look particularly weak, and the winner-takes-all approach indicates that there are voxels that are most strongly activated for the nose compared to all other facial parts. This could be explained in more detail. Related to this point, did the authors test for nose remapping in amputees and one-handers?

Following this comment, we have re-ran all univariate analyses to include the nose condition, and updated all results throughout the main text, supplemental results and related figures. We initially chose to exclude the nose from univariate analyses because many participants could not move it. RSA is less sensitive to noise (due to cross-validation) and to selectivity/topography (two issues we had with the nose condition) and for these reasons we initially decided to keep the nose in the RSA analysis and exclude it from the univariate analyses. But we agree that being consistent avoids unnecessary confusion, and adding the nose condition into all analyses did not change the univariate results, apart from a now significant group x hemisphere interaction for the CoG of the tongue when comparing amputees and controls, better matching the trends for greater surface coverage in the deprived hand ROI of amputees.

We also performed and added in the Supplementary section the univariate results for the nose (Appendix Figure 1). We only shortly mention this result in the main text (lines 324-330) since we feel like the Results section is already pretty dense with results, and this additional result does not change our interpretation and conclusions.

What happened with the thumb movement imagery in the one-handers? Suppl. Figure 2 shows the respective effects for amputees and controls. Did the one-handers fail to show significant effects? It might be interesting to compare the one-handers' imagery effects with the execution effects in amputees and controls, even if putative differences might be less straightforward to interpret (might be due to representational differences or different paradigms).

In short – we cannot answer this question. To clarify, this condition was present solely to fill the experimental gap so that the protocols were matched as closely as possible across groups and we never intended to analyse this condition in one-handers. While we agree it could have been interesting to study how one-handers relate to their missing hand, there are different kinds of motor imagery (e.g., visual, kinaesthetic) and without dedicated research tools it would have been tricky for us to monitor individual one-handers strategies for imagining movement of a limb they never had. Consequently, we gave only broad instructions for this condition, that one-handers could interpret in various ways (visual, motor imagery or even prosthesis movements). It thus does not make much sense for us to look at it, and will not be meaningful to compare to the amputees group where we very explicitly instructed participants to move their phantom hand. Appendix Figure 3 addresses the question of persistent representation in amputees, which is highly relevant for this paper. To reduce the discrepancy, we removed the analysis including the phantom hand (see next comment). And we added this explanation in the methods (lines 645-647).

Related to this point, the face-thumb distance values in Figure 5 are only shown for the controls and amputees. For the reader, the reason is not immediately clear, so perhaps this could be clarified in the figure caption.

Upon reflection (and thanks to the reviewers’ comments), we decided to remove entirely the hand-face dissimilarity analysis, which does not directly relate to the question of remapping (but rather of shared representation), in addition to making the paper unbalanced (as explained above, we were unable to run this analysis in one-handers). We will now feature this analysis in another paper that appears more appropriate in the context of referred sensations in amputees (Amoruso et al., 2022 MedRxiv).

There are differences between the forehead clusters found in the winner-takes-all analysis (Figure 1) and the group consistency maps; the latter showing a strong overlap with the hand ROI. For the lips and tongue, the maps appear much more consistent. What could be the reason for this discrepancy, and how can the overlap between hand ROI and forehead be interpreted?

We apologise for any confusion. Figure 1 represents the thresholded group-level maps (vs rest, *not* winner-takes-all) within our initial (and very broad) sensorimotor mask, whereas the winner-takes-all inter-subject consistency maps are displayed in the following figures (2 to 4). The strong overlap with the hand ROI observed for the forehead is due to the use of an ROI including both the face and hand regions, where hand activity (not available for all groups) was not included. The logic behind this choice was that including the hand ROI would ease the detection of remapping (into the hand ROI). Given the winner-takes-all procedure (which included minimal thresholding), and the fact that the forehead is the immediate neighbour of the hand area, the slightly stronger forehead activity dominated the hand area. But when more stringent thresholding is introduced (as in Figure 1), the forehead activity is insufficient to pass the threshold. To add clarity to the activity presented in Figure 1, we have visualised the pre-thresholding mask of the pre- and post-central gyrus (defined using the Harvard Cortical Atlas in dark grey; see line 172), as well as the hand and face ROIs used in the subsequent winner-takes-all analysis, and amended Figure legends to state that the group-level activity visualised is versus rest (line 167) and winner-takes-all maps done within the combined hand+face ROI (line 199-200). We have also updated the figure legend of Figure 2 (lines 202-204) to explicitly acknowledge these differences: “Please note that the individual-participant winner-takes-all maps are minimally thresholded, and thus produce an inherently different spatial distribution relative to the group contrast maps presented in Figure 1”.

When reporting the weak but significant result of reduced forehead coverage in amputees, the authors could also refer to the BF=1.5, which suggests only anecdotal evidence.

We agree with the reviewer’s above recommendation, however we initially decided to provide Bayes Factors only in the absence of effects, to show whether we had enough evidence in support for the null hypothesis (see lines 877-879: “We reported the corresponding Bayes Factor (BF10), defined as the relative support for the alternative hypothesis, for non-significant interactions and post-hoc comparisons.”).

The authors did not remove outliers, which is justified, and the authors checked that the significance and direction of the results did not change if identified outliers were removed. However, could the authors indicate the number of outliers?

We have added the number of outliers to the ‘Statistical Analyses’ section (lines 863-864, 877, 889, 894, 896) of the Materials and methods. As detailed, there were very few outliers (up to one per condition/group).

Reviewer #2 (Recommendations for the authors):After amputation, the deafferented limb representation in the somatosensory cortex is activated by stimulation of other body parts. A common belief is that the lower face, including the lips, preferentially "invades" deafferented cortex due to its proximity to cortex. In the present study, this hypothesis is tested by mapping the somatosensory cortex using fMRI as amputees, congenital one-handers, and controls moved their forehead, nose, lips or tongue. First, they found that, unlike its counterpart in monkeys, the representation of the face in the somatosensory cortex is right-side up, with the forehead most medial (and abutting the hand) and the lips most lateral. Second, there was little evidence of "reorganization" of the deafferented cortex in amputees, even when tested with movements across the entire face rather than only the lips. Third, congenital one-handers showed significant reorganization of deafferented cortex, characterized principally by the invasion of the lower face, in contrast to predictions from the hypothesis that proximity was the driving factor. Fourth, there was no relationship between phantom limb pain reports and reorganization.As a non-expert in fMRI, I cannot evaluate the methodology. That being said, I am not convinced that the current consensus is that the representation of the face in humans is flipped compared to that of monkeys. Indeed, the overwhelming majority of somatosensory homunculi I have seen for humans has the face right side up. My sense is that the fMRI studies that found an inverted (monkey-like) face representation contradict the consensus.

Thank you for point this out. As we tried to emphasise in the introduction, very few neuroimaging studies actually investigated face somatotopy in humans, with inconsistent results. We agree the default consensus tends to be dominated by the up-right depiction of Penfield’s homunculus (recently replicated by Roux et al., 2018). However, due to methodological and practical constraints, alignment across subjects in the case of intracortical recordings is usually difficult to achieve, and thus makes it difficult to assess the consistency in topographical organisation. Moreover, *previous imaging studies did not manage to convincingly support Penfield’s homunculus*. For these two key reasons, the spatial orientation of the human facial homunculus is still debated. A further limiting factor of previous studies in humans is that the vast majority of human studies investigating face (re)mapping in humans focused solely on the lip representation, using the cortical proximity hypothesis to interpret their results. Consequently, as we highlight above in our response to the Editor, there is a wide-spread and false representation in the human literature of the lips neighbouring the hand area.

To account for the reviewer’s critic and convey some of this context, we changed our title from: *Reassessing face topography in primary somatosensory cortex and remapping following hand loss*; to: *Complex pattern of facial remapping in somatosensory cortex following congenital but not acquired hand loss*. This was done to de-emphasise the novelty of face topography relative to our other findings.

We also rewrote our introduction (lines 79-94) as follows:

“The research focus on lip cortical remapping in amputees is based on the assumption that the lips neighbour the hand representation. However, this assumption goes against the classical upright orientation of the face in S1^26–30^, as first depicted in Penfield’s Homunculus and in later intracortical recordings and stimulation studies^26–29^, with the upper-face (i.e., forehead) bordering the hand area. In contrast, neuroimaging studies in humans studying face topography provided contradictory evidence for the past 30 years. While a few neuroimaging studies provided partial evidence in support of the traditional upright face organisation^31^, other studies supported the inverted (or ‘upside-down’) somatotopic organisation of the face, similar to that of non-human primates^32,33^. Other studies suggested a segmental organisation^34^, or even a lack of somatotopic organisation^35–37^, whereas some studies provided inconclusive or incomplete results^38–41^. Together, the available evidence does not successfully converge on face topography in humans. In line with the upright organisation originally suggested by Penfield, recent work reported that the shift in the lip representation towards the missing hand in amputees was minimal^42,43^, and likely to reside within the face area itself. Surprisingly, there is currently no research that considers the representation of other facial parts, in particular the upper-face (e.g., the forehead), in relation to plasticity or PLP.”

We also updated the discussion accordingly (lines 456, 468-476, 489-491).

Similarly, it is not clear to me how the observations (1) of limited reorganization in amputees, (2) of significant reorganization in congenital one-handers, and (3) of the lack of relationship between PLP and reorganization is novel given the previous work by this group. Perhaps the authors could more clearly articulate the novelty of these results compared to their previous findings.

Thank you for giving us the opportunity to clarify on this important point. The novelty of these results can be summarised as follow:

(1)Conceptually, it is crucial for us to understand if deprivation-triggered plasticity is constrained by the local neighbourhood, because this can give us clues regarding the mechanisms driving the remapping. We provide strong topographic evidence about the face orientation in controls, amputees and one-handers.(2)The vast majority of previous research on brain plasticity following hand loss (both congenital and acquired) in humans has exclusively focused on the lower face, and lips in particular. We provide systematic evidence for stable organisation and remapping of the neighbouring upper face, as well as the lower face. We also study topographic representation of the tongue (and nose) for the first time.(3)The vast majority of previous research on brain remapping following hand loss (both congenital and acquired, neuroimaging and electrophysiological) was focused on univariate activity measures, such as the spatial spread of units showing a similar feature preference, or the average activity level across individual units. We are going beyond remapping by using RSA, which allows us to ask not only if new information is available in the deprived cortex (as well as the native face area), but also whether this new information is structured consistently across individuals and groups. We show that representational content is enhanced in the deprived cortex one-handers whereas it is stable in amputees relative to controls (and to their intact hand region).(4)Based on previous studies, the assumption was that reorganisation in congenital one-handers was relatively unspecific, affecting all tested body parts. Here, we provide evidence for a more complex pattern of remapping, with the forehead representation seemingly moving out of the missing hand region (and the nose representation being tentatively similar to controls). That is, we show not just “invasion” but also a shift of the neighbour away from the hand area which has never been documented (or in fact suggested).(5)Using Bayesian analyses we provide definitive evidence against a relationship between PLP and forehead remapping, providing first and conclusive evidence against the remapping hypothesis, based on cortical neighbourhood.

Our inclination is not to add a summary paragraph of these points in our discussion, as it feels too promotional. Instead, we have re-written large sections of the introduction and discussion to better emphasise each of these points separately throughout the text, where the context is most appropriate. Given the public review strategy taken by *eLife*, the novelty summary provided above will be available for any interested reader, as part of the public review process. However, should the reviewer feel that a novelty summary paragraph is required (or an emphasis on any of the points summarised above), we will be happy to revise the manuscript accordingly.

Finally, Jon Kaas and colleagues (notably Niraj Jain) have provided evidence in experiments with monkeys that much of the observed reorganization in the somatosensory cortex is inherited from plasticity in the brain stem. Jain did not find an increased propensity for axons to cross the septum between face and hand representations after (simulated) amputation. From this perspective, the relevant proximity would be that of the cuneate and trigeminal nuclei and it would be critical to map out the somatotopic organization of the trigeminal and cuneate nuclei to test hypotheses about the role of proximity in this remapping.

Thank you for highlighting this very relevant point, which we are well aware of. We fully agree with the reviewer that this is an important goal for future study, but functional imaging of the brainstem in humans is particularly challenging and would require ultra high field imaging (7T) and specialised equipment. We have encountered much local resistance due to hypothetical issues for MRI safety for scanning amputees in this higher field strength, meaning we are unable to carry out this research ourselves. Our former lab member Sanne Kikkert, who is now running her independent research programme in Zurich, has been working towards this goal for the past 4 years. So we can say with confidence that this aim is well beyond the scope of the current study. In response to your comment, we mentioned this potential mechanism in the introduction (lines 98-101), we ensured that we only referred to “cortical proximity” throughout our manuscript, and we circle back to this important point in the discussion.

Lines 538-542: “Moreover, even if the remapping we observed here goes against the theory of *cortical* proximity, it can still arise from representational proximity at the subcortical level, in particular at the brainstem level^44,45^. While challenging in humans, mapping both the cuneate and trigeminal nuclei would be critical to provide a more complete picture regarding the role of proximity in remapping.”

A few comments:Which cortical fields do the authors refer to as primary somatosensory cortex? Brodmann's areas 3a, 3b, 1 and 2?On line 322, I suggest using "completeness" rather than "completion."On line 372, I suggest using "critique" rather than "limitation."

We thank the reviewer for the above minor comments and can confirm that our primary somatosensory cortex refers to Brodmann’s areas 1, 2, 3a and 3b (stated in Regions of Interest (ROI) definition section, line 714, ‘Materials and methods’). We have also changed the suggested words now on line 438 and line 480 based on the reviewer’s recommendation.

Reviewer #3 (Recommendations for the authors):In their study, the authors set up to challenge the long-held claim that cortical remapping in the somatosensory cortex in hand deprived cortical territories follows somatotopic proximity (the hand region gets invaded by cortical neighbors) as classically assumed. In contrast to this claim, the authors suggest that remapping may not follow cortical proximity but instead functional rules as to how the effector is used. Their data indeed suggest that the deprived hand area is not invaded by the forefront which is the cortical neighbor but instead by the lips which may compensate for hand loss in manipulating objects. Interestingly the authors suggest this is mostly the case for one-handers but not in amputees for who the reorganization seems more limited in general (but see my comments below on this last point).This is a remarkably ambitious study that has been skilfully executed on a strong number of participants in each group. The complementarity of state-of-the-art uni- and multi-variate analyses are in the service of the research question, and the paper is clearly written. The main contribution of this paper, relative to previous studies including those of the same group, resides in the mapping of multiple face parts all at once in the three groups.

We are grateful to the reviewer for appreciating the immense effort that this study involved.

In the winner takes all approach, the authors only include 3 face parts but exclude from the analyses the nose and the thumb. I am not fully convinced by the rationale for not including nose in univariate analyses – because it does not trigger reliable activity – while keeping it for representational similarity analyses. I think it would be better to include the nose in all analyses or demonstrate this condition is indeed "noisy" and then remove it from all the analyses. Indeed, if the activity triggered by nose movement is unreliable, it should also affect multivariate.

Following this comment, we re-ran all univariate analyses to include the nose, and updated throughout the main text and supplemental results and related figures. In short, adding the nose did not change the univariate results, apart from a now significant group x hemisphere interaction for the CoG of the tongue when comparing amputees and controls, matching better the trends for greater surface coverage in the deprived hand ROI of amputees. Full details are provided in our response to Reviewer 1 above.

The rationale for not including the hand is maybe more convincing as it seems to induce activity in both controls and amputees but not in one-handers. First, it would be great to visualize this effect, at least as supplemental material to support the decision. Then, this brings the interesting possibility that enhanced invasion of hand territory by lips in one-handers might link to the possibility to observe hand-related activity in the presupposed hand region in this population. Maybe the authors may consider linking these.

Thank you for this comment. As we explain in our response to Reviewer 1 above, we did not intent the thumb condition in one-handers for analysis, as the task given to one-handers (imagine moving a body part you never had before) is inherently different to that given to the other groups (move – or at least attempt to move – your (phantom) hand). As such, we could not pursuit the analysis suggested by the reviewer here. To reduce the discrepancy and following Reviewer 1’s advice, we decided to remove the hand-face dissimilarity analysis which we included in our original manuscript, and might have sparked some of this interest. Upon reflection we agreed that this specific analysis does not directly relate to the question of remapping (but rather of shared representation), in addition to making the paper unbalanced. We will now feature this analysis in another paper that appears more appropriate in the context of referred sensations in amputees (Amoruso et al., 2022 MedRxiv).

The use of the geodesic distance between the center of gravity in the Winner Take All (WTA) maps between each movement and a predefined cortical anchor is clever. More details about how the Center Of Gravity (COG) was computed on spatially disparate regions might deserve more explanations, however.

We are happy to provide more detail on this analysis, which weights the CoG based on the clusters size (using the workbench command -metric-weighted-stats). Let’s consider the example in Author response image 1 for a single control participant, where each CoG is measured either without weighting (yellow vertices) or with cluster weighting (forehead CoG=red, lip CoG=dark blue, tongue CoG=dark red). When the movement produces a single cluster of activity (the lips in the non-dominant hemisphere, shown in blue), the CoG’s location was identical for both weighted (red) and unweighted (yellow) calculations. But other movements, such as the tongue (green), produced one large cluster (at the lateral end), with a few more disparate smaller clusters more medially. In this case, the larger cluster of maximal activity is weighted to a greater extent than the smaller clusters in the CoG calculation, meaning the CoG is slightly skewed towards it (dark red), relative to the smaller clusters.

**Author response image 1. sa2fig1:** Centre-of-gravity calculation, weighted and unweighted by cluster size, in an example control participant. Here the winner-takes-all output for each facial movement (forehead=red, lips=blue, tongue=green) was used to calculate the centre-of-gravity (CoG) at the individual-level in both the dominant (left-hand side) and non-dominant (right-hand side) hemisphere, weighted by cluster size (forehead CoG=red, lip CoG=dark blue, tongue CoG=dark red), compared to an unweighted calculation (denoted by yellow dots within each movements’ winner-takes-all output).

This is now explained in the methods (lines 759-764) as follows:

“To assess possible shifts in facial representations towards the hand area, the centre-of-gravity (CoG) of each face-winner map was calculated in each hemisphere. The CoG was weighted by cluster size meaning that in the event of multiple clusters contributing to the calculation of a single CoG for a face-winner map, the voxels in the larger cluster are overweighted relative to those in the smaller clusters. The geodesic cortical distance between each movement’s CoG and a predefined cortical anchor was computed.”

Moreover, imagine that for some reason the forefront region extends both dorsally and ventrally in a specific population (eg amputees), the COG would stay unaffected but the overlap between hand and forefront would increase. The analyses on the surface area within hand ROI for lips and forehead nicely complement the WTA analyses and suggest higher overlap for lips and lower overlap for forehead but none of the maps or graphs presented clearly show those results – maybe the authors could consider adding a figure clearly highlighting that there is indeed more lip activity IN the hand region.

We agree with you on this limitation of the CoG and this is why we interpret all cortical distances analyses in tandem with the laterality indices. The laterality indices correspond to the proportion of surface area in the hand region for a given face part in the winner-maps.

Nevertheless, to further convince the Reviewer, we extracted activity levels (β values) within the hand region of congenitals and controls, and we ran (as for CoGs) a mixed ANOVA with the factors Hemisphere (deprived x intact) and Group (controls x one-handers).

As expected from the laterality indices obtained for the Lips, we found a significant group x hemisphere interaction (*F_(1,41_*_)_=4.52, *p*=0.040, *n^2^_p_*=0.099), arising from enhanced activity in the deprived hand region in one-handers compared to the non-dominant hand region in controls (*t_(41)_*=-2.674*, p*=0.011) and to the intact hand region in one-handers (*t_(41)_*=-3.028*, p*=0.004).

Since this kind of analysis was the focus of previous studies (from which we are trying to get away) and since it is redundant with the proportion of face-winner surface coverage in the hand region, we decided not to include it in the paper. But we could add it as a Supplementary result if the Reviewer believes this strengthens our interpretation.

In addition to overlap analyses between hand and other body parts, the authors may also want to consider doing some Jaccard similarity analyses between the maps of the 3 groups to support the idea that amputees are more alike controls than one-handers in their topographic activity, which again does not appear clear from the figures.

We thank the reviewers for this clever suggestion. We now include the Jaccard similarity analysis, which quantified the degree of similarity (0=no overlap between maps; 1=fully overlapping) between winner-takes-all maps (which included the nose; akin to the revised univariate results) across groups. For each face part/amputee, the similarity with the 22 controls and 21 one-handers respectively was averaged. We utilised a linear mixed model which included fixed factors of Group (One-handers x Controls), Movement (Forehead x Nose x Lips x Tongue) and Hemisphere (Intact x Deprived) on Jaccard similarity values (similar to what we used for the RSA analysis). A random effect of participant, as well as covariates of ages, were also included in the model.

Results showed a significant group x hemisphere interaction (*F_(240.0)_*=7.70, *p*=0.006; controlled for age; Figure 5), indicating that amputees’ maps showed different similarity values to controls’ and one-handers’ depending on the hemisphere. Post-hoc comparisons (corrected α=0.025; uncorrected *p*-values reported) revealed significantly higher similarity to controls’ than to one-handers’ maps in the deprived hemisphere (*t_(240)_*=-3.892, *p*<.001). Amputees’ maps also showed higher similarity to controls’ maps in the deprived relative to the intact hemisphere (*t_(240)_*=2.991, *p*=0.003). Amputees, therefore, displayed greater similarity of facial somatotopy in the deprived hemisphere to controls, suggesting again fewer evidence for cortical remapping in amputees.

We added these results at the end of the univariate analyses (lines 334-350) and in the discussion (lines 463-464 and 496-499).

This brings to another concern I have related to the claim that the change in the cortical organization they observe is mostly observed in one-handers. It seems that most of this conclusion relies on the fact that some effects are observed in one-handers but not in amputees when compared to controls, however, no direct comparisons are done between amputees and one-handers so we may be in an erroneous inference about the interaction when this is actually not tested (Nieuwenhuis, 11). For instance, the shift away from the hand/face border of the forehead is also (mildly) significant in amputees (as observed more strongly in one-handers) so the conclusion (eg from the subtitle of the Results section) that it is specific to one-hander might not fully be supported by the data. Similar to the invasion of the hand territory from the lips which is significant in amputees in terms of surface area. All together this calls for toning down the idea that plasticity is restricted to congenital deprivation (eg last sentence of the abstract). Even if numerically stronger, if I am not wrong, there are no stats showing remapping is indeed stronger in one-handers than in amputees and actually, amputees show significant effects when compared to controls along the lines as those shown (even if more strongly) in one-handers.

Thank you for this very important comment. We fully agree – the RSA across-groups comparison is highly informative but insufficient to support our claims. We did not compare the groups directly to avoid multiple comparisons (both for statistical reasons and to manage the size of the Results section). But the reviewer’s suggestion to perform a Jaccard similarity analysis complements very nicely the univariate and multivariate results and allows for a direct (and statistically lean) comparison between groups, to assess whether amputees are more similar to controls or to congenital one-handers, taking into account all aspects of their maps (both spatial location/CoG and surface coverage). We added the Jaccard analysis to the main text, at the end of the univariate results (lines 334-384). The Jaccard analysis suggests that amputees’ maps in the deprived hemisphere were more similar to the maps of controls than to the ones of congenital one-handers. This allowed us to obtain significant statistical results to support the claim that remapping is indeed stronger in one-handers than in amputees (lines 345-350). We also compared both amputees and one-handers to the control group. In line with our univariate results, this revealed that the only face part for which controls were more similar to one-handers than to amputees was the tongue (lines 378-380). And that the forehead remapping observed at the univariate level in amputees (surface area), is likely to arise from differences in the intact hemisphere (lines 380-382).

Finally, we also added the post-hoc statistics comparing amputees to congenitals in the RSA analysis (lines 424-426): “While facial information in the deprived hand area was increased in one-handers compared with amputees, this effect did not survive our correction for multiple comparisons (*t*_(70.7)_=-2.117, *p*=0.038).”

Regarding the univariate results mentioned by the reviewer, we would like to emphasise that we had no significant effect for the lips in amputees, though we agree the surface area appears in between controls and one-handers. But this laterality index was not different from zero. This test is now added lines 189-190. Regarding the forehead, we fully agree with the Reviewer, and we adjusted the subtitle accordingly (lines 240-241). For consistency, we also added the t-test vs zero for the forehead surface area (non-significant, lines 250-252).

Also, maybe the authors could explore whether there is actually a link between the number of years without hand and the remapping effects.

To address this question, we explored our data using a correlation analysis. The only body part who showed some suggestive remapping effects was the tongue, and so we explored whether we could find a relationship (Pearson’s correlation) between years since amputation and the laterality index of the Tongue in amputees (r = 0.007, p=0.980, 95% CI [-0.475, 0.475]). We also explored amputees’ global Jaccard similarity values to controls in the deprived hemisphere (r = -0.010, p=0.970, 95% CI [-0.488, 0.473]), and could not find any relationship. Considering there was no strong remapping effect to explain, we find this result too exploratory to include in our manuscript.

One hypothesis generated by the data is that lips remap in the deprived hand area because lips serve compensatory functions. Actually, also in controls, lips and hands can be used to manipulate objects, in contrast to the forehead. One may thus wonder if the preferential presence of lips in the hand region is not latent even in controls as they both link in functions?

We agree with the reviewer’s reasoning, and we think that the distributed representational content we recently found in two-handers (Muret et al., 2022) provides a first hint in this direction. It is worth noting that in that previous publication we did not find differences across face parts in the activity levels obtained in the hand region, except for slightly more negative values for the tongue. But we do think that such latent information is likely to provide a “scaffolding” for remapping. While the design of our face task does not allow to assess information content for each face part (as done for the lips in Muret et al., 2022), this should be further investigated in follow-up studies.

We added a sentence in the discussion to highlight this interesting notion:

Lines 555-558: “Together with the recent evidence that lip information content is already significant in the hand area of two-handed participants (Muret et al., 2022), compensatory behaviour since developmental stages might further uncover (and even potentiate) this underlying latent activity.”

1) The authors should re-assess their conclusion that remapping is mostly present in one-handers than amputees in absence of a convincing statistical demonstration that it is actually the case. The amputees show similar effects or trends for similar remapping as those observed in one-handers. maybe one source of this lower reliability in the effects of amputees links to the lower number of participants and their higher variability in ethology (eg age of amputation etc…). Maybe the authors could explore whether there is actually a link between the number of years without hand and the remapping effects.

We would like to emphasise that our Bayesian statistics help address this issue when supportive of the null hypothesis (i.e., lack of interaction for the lip CoG: *BF_10_*=0.297). But we also fully agree that a more direct comparison between amputees and one-handers was necessary. As we detailed above, we now provide analysis to address both univariate and multivariate group differences, performed with the two most sensitive analyses, Jaccard analysis and RSA.

In the Jaccard analysis, amputees showed significantly higher similarity to controls, and their global similarity to controls was not related to covariates such as years since amputation. We also show in *Figure 5 —figure supplement 1* comparable intra-group consistency in amputees relative to one-handers. We believe this rules out the reviewer’s concern that the amputees data is more noisy/variable due to amputation-related covariates.

2) The authors should be coherent in the body part they include for their univariate and multivariate analyses. I let them decide whether they think the nose condition was too noisy (eg because people could not follow instructions to move the nose) or not. If they think the condition however did not elicit reliable activity, this should affect both uni and multivariate analyses.

We have addressed this comment in the most straight forward way we could – re-running all analyses to include the nose and update all of the study results accordingly. Despite slight quantitative changes, the overall pattern of the results was relatively unchanged in comparison to our original analysis.

3) I recommend the authors show the map elicited by hand movement in the three groups, support their claim that this is only reliably observed in controls and amputees but not one-handers; and maybe discuss (or even better do some analyses on that, eg correlation) the possibility that the preservation of the hand map is potentially a factor influencing remapping (eg do the amputees with the most salient remaining hand maps are those showing less remapping?).

We thank the reviewer for their interesting suggestion that the preservation of the hand map may influence reported (lack of) remapping. To explore this within our data, and from the recommendation of the reviewer, we looked at the correlation between the percentage of surface area coverage for maximal phantom hand activity in the hand ROI (similar to that shown in Appendix Figure 3 in standard space) to the laterality indices of the tongue (the one facial part in amputees to tentatively indicate facial remapping). When doing so we find a non-significant correlation (kendall's tau=0.121, p=0.281, BF10=0.366), along with an inconclusive Bayes Factor, suggesting that preservation of phantom hand activity may not be a deciding factor of facial remapping within this dataset. Considering there was no strong remapping effect to explain, we find this result too exploratory to include in our manuscript. But we would happily add it to the Supplements if advised by the Reviewer.

We therefore continued with an alternative approach that might make more intuitive sense – do we have more stability, thus more control-like maps, if amputees display more (preserved) hand representation? To address this question, we ran a Pearson correlation between the global Jaccard similarity values of amputees relative to controls and the percentage of surface area coverage for maximal phantom hand activity in the hand ROI (similar to that shown in Appendix Figure 3 in standard space). We found here again a non-significant correlation (r = 0.317, p=0.215, 95% CI [-0.193, 0.692]).

**Author response image 3. sa2fig3:** 

4) The authors should clarify in their figure where do they observe that the lip increases activity WITHIN the hand region in one-handers.

The laterality indices are clearly showing this, since it corresponds to the proportion of surface area coverage IN the hand region, for the winner-maps of each face part (thus winning over the other face parts). See above for a similar analysis on activity levels (β values) in the hand region.

5) The fact that the univariate results of Figure 1 do not follow well those presented in the WTA maps deserves some consideration. Indeed, what seems striking in Figure 1 is that the forehead activity extends more dorsally in one-handers, approaching the hand region, which contradicts conclusions from WTA. Also, it might be informative to have an outline of the brain mask (in all figures) of the S1 ROI used for constraining spatially the univariate analyses.

We apologise for any confusion. As we detail in our response to Reviewer 1 above, the activity presented in Figure 1 reflects each facial movement contrasted to rest, mainly localised outside, and posterior to, the hand ROI (outlined in Figure 1 in purple). The WTA maps, summarised in the consistency maps in Figures 2-4, reflect how consistent are the (minimally thresholded) WTA maps. If we used a higher threshold, much of the upper face WTA map would disappear.

6) In Figure 5 the thumb is not represented in one-handers. It would be interesting for the reader to see that indeed the instruction to move the thumb did not elicit a reliable activity map and what was the criteria to selectively remove it in one-handers aside from the fact they don't experience fantom sensation. It is also unclear why the thumb is not displayed in the MDS while it's computed in the RDM of controls and amputees.

Upon reflection (and thanks to the reviewers’ comments), we decided to remove entirely the hand-face dissimilarity analysis, which does not directly relate to the question of remapping (but rather of shared representation), in addition to making the paper unbalanced. We will now feature this analysis in another paper that appears more appropriate in the context of referred sensations in amputees (Amoruso et al., 2022 MedRxiv).

7) Maybe discuss further the potential (latent) link between hand and lips as they both can be used to manipulate objects even in controls.

We added a sentence in the discussion to that extent:

Lines 555-558: “Together with the recent evidence that lip information content is already significant in the hand area of two-handed participants (Muret et al., 2022), compensatory behaviour since developmental stages might further uncover (and even potentiate) this underlying latent activity.”